# Kinetic and data-driven modeling of pancreatic β-cell central carbon metabolism and insulin secretion

**Patrick E. Gelbach**[1], **Dongqing Zheng**[2], **Scott E. Fraser**[3], **Kate L. White**[4], **Nicholas A. Graham**[2], **Stacey D. Finley**[1,2,5]*

1 Department of Biomedical Engineering, USC, Los Angeles, California, United States of America, 2 Mork Family Department of Chemical Engineering and Materials Science, USC, Los Angeles, California, United States of America, 3 Translational Imaging Center, University of Southern California, Los Angeles, California, United States of America, 4 Departments of Biological Sciences and Chemistry, Bridge Institute, USC Michelson Center, USC, Los Angeles, California, United States of America, 5 Department of Quantitative and Computational Biology, USC, Los Angeles, California, United States of America

* sfinley@usc.edu

**Data Availability Statement:** All relevant data are within the manuscript and its Supporting Information files. Model scripts are available at:

## Abstract

Pancreatic β-cells respond to increased extracellular glucose levels by initiating a metabolic shift. That change in metabolism is part of the process of glucose-stimulated insulin secretion and is of particular interest in the context of diabetes. However, we do not fully understand how the coordinated changes in metabolic pathways and metabolite products influence insulin secretion. In this work, we apply systems biology approaches to develop a detailed kinetic model of the intracellular central carbon metabolic pathways in pancreatic β-cells upon stimulation with high levels of glucose. The model is calibrated to published metabolomics datasets for the INS1 823/13 cell line, accurately capturing the measured metabolite fold-changes. We first employed the calibrated mechanistic model to estimate the stimulated cell's fluxome. We then used the predicted network fluxes in a data-driven approach to build a partial least squares regression model. By developing the combined kinetic and data-driven modeling framework, we gain insights into the link between β-cell metabolism and glucose-stimulated insulin secretion. The combined modeling framework was used to predict the effects of common anti-diabetic pharmacological interventions on metabolite levels, flux through the metabolic network, and insulin secretion. Our simulations reveal targets that can be modulated to enhance insulin secretion. The model is a promising tool to contextualize and extend the usefulness of metabolomics data and to predict dynamics and metabolite levels that are difficult to measure *in vitro*. In addition, the modeling framework can be applied to identify, explain, and assess novel and clinically-relevant interventions that may be particularly valuable in diabetes treatment.

## Author summary

Diabetes is among the most common chronic illnesses, occurring when the β-cells in the pancreas are unable to produce enough insulin to properly manage the body's blood sugar

https://github.com/FinleyLabUSC/
PancreaticBetaCellMetabolism.

**Funding:** SDF, NAG, KW, and SEF received support from the USC Pancreatic Beta Cell Consortium. SDF received support from the NIH National Cancer Institute (1U01CA232137). The funders had no role in study design, data collection and analysis, decision to publish, or preparation of the manuscript.

**Competing interests:** The authors have declared that no competing interests exist.

levels. β-cells metabolize nutrients to produce energy needed for insulin secretion in response to high glucose, and there is a potential to harness β-cell metabolism for treating diabetes. However, β-cell metabolism is not fully characterized. We have developed a computational modeling framework to better understand the relationship between cellular metabolism and insulin production in the pancreatic β-cell. With this modeling framework, we are able to simulate metabolic perturbations, such as the knockdown of the activity of a metabolic enzyme, and predict the effect on the metabolic network and on insulin production. This work can therefore be applied to investigate, in a time- and cost-efficient manner, β-cell metabolism and predict effective therapies that target the cell's metabolic network.

## 1. Introduction

Pancreatic β-cells, the predominant cell type in the pancreatic islets of Langerhans, respond to and tightly regulate the body's blood glucose levels through insulin secretion. The process of glucose-stimulated insulin secretion (GSIS) is heavily dependent on the cells' intracellular metabolism [1, 2]. Upon stimulation with high glucose levels, glucose is transported into the cell, causing an increase in glycolysis and oxidative phosphorylation, which lead to an increase in the cellular ATP/ADP ratio [3]. Increased ATP causes the closure of potassium ($K^+$) channels and the opening of calcium ($Ca^{2+}$) channels, which promote the release of insulin. In order to accomplish this cascade of events, the β-cell has several key glucose-sensing metabolic steps that are widely considered to be vital to insulin secretion, including a specialized glucose transporter and the glucokinase reaction [4, 5]. However, there are many additional pathways, metabolites, and reactions that are purported to be impactful in insulin secretion, depending on the context [6–10]. Additionally, the way coordinated changes in metabolic pathways and resulting metabolite pools influence insulin secretion is not fully understood. For these reasons, there is a need to study pancreatic β-cell metabolism at a systems-level, identifying how sets of metabolic pathways work together to cause the observed biological properties of insulin-secreting pancreatic β-cells.

Given that appropriate secretion of insulin is vital to the successful maintenance of blood glucose homeostasis, an impaired metabolic state of the β-cell is closely linked to disease progression [11]. Lowered insulin secretion is correlated with the emergence and severity of Type 2 diabetes [12, 13]. There is significant value in developing a deeper understanding of β-cell metabolic activity to determine underlying biological processes driving disease progression and to find novel potential mechanisms to treat the disease. Mass spectrometry-based metabolomics, which enables quantitative measurements of cellular metabolites, has emerged as a way to analyze the cell's metabolic condition and thereby elucidate metabolic processes, in both healthy and diseased conditions. Because measurements of metabolite pool sizes alone do not give holistic insight into cellular behavior, their utility may be extended by integrating the data into frameworks that predict transient dynamics, thus linking experimental measurements with phenotypic understanding [14]. Computational modeling of metabolism is one such technique that can leverage metabolomics data to predict dynamic behavior. Especially because the GSIS system is inherently time-dependent, it is valuable to use nonlinear ordinary differential equation (ODE) models, trained and refined with metabolomics data, to further our understanding of the mechanisms driving insulin secretion.

Several mathematical models have been developed to understand the relationship between β-cell metabolism and insulin production. Topp et al. developed a simple three-equation

model describing the relationships between glucose, insulin, and β-cell mass [15]. Modeling work by Bertram et el. focused primarily on ATP synthesis from pyruvate [16]. Magnus and Keizer studied the underlying mechanisms driving calcium cycling in β-cells, while Yugi and Tomita focused on mitochondrial metabolic activity [17–19]. Fridyland focused on the link between cellular metabolism and energetic processes such as the maintenance of a mitochondrial membrane potential [20]. Jiang et al. built upon many of these works to develop a detailed kinetic model of glucose-stimulated metabolism in β-cells [21]. However, the Jiang model lacked several glycolytic metabolites and pathways that may contribute to the activity of pancreatic β-cells, and it did not link insulin secretion to the modeled metabolic processes. Thus, while computational modeling has been used, there is a current need for greater understanding of core central carbon metabolic pathways and how their dynamics correlate to insulin secretion.

To address this gap in knowledge, we develop a kinetic model of pancreatic β-cell intracellular metabolism, including key β-cell-specific metabolic pathways. We refine and train the model with published *in vitro* metabolomics data and assess the impact of metabolic perturbations on the entire network. We also pair predictions from the kinetic model with linear regression analysis to link metabolic processes with insulin secretion. Our integrative modeling approach is therefore a valuable tool to understand the dynamics of GSIS and may inform future research aimed at treating β-cell dysfunction.

## 2. Materials and methods

### 2.1 Model structure and metabolic pathways

We constructed a kinetic model of the central carbon metabolism of the INS1 832/13 pancreatic β-cell line by building upon previously published models of intracellular metabolism (**Fig 1**) [21–23]. The model consists of 56 metabolites and 65 enzyme-catalyzed metabolic reactions in six primary metabolic pathways: glycolysis, glutaminolysis, the pentose phosphate pathway (PPP), the tricarboxylic acid (TCA) cycle, the polyol pathway, and electron transport chain (ETC). These reactions occur in two cellular sub-compartments (cytosol and mitochondria). By including these central carbon metabolic pathways, the model is significantly more detailed and expansive than other published models of pancreatic β-cell metabolism. The model is represented as a series of nonlinear ODEs, characterized with Michaelis-Menten or bi-bi reaction kinetics, that describe how the concentrations of intracellular metabolites evolve over time in a pancreatic β-cell [24, 25]. Thus, there is a single ODE for each metabolite included in the model. Metabolites that are found in both cytosol and mitochondria have separate equations, allowing for the comparison of concentrations between the two cellular sub-compartments. The ODEs are implemented in MATLAB and solved with the built-in *ode15s* differential equation solver [26]

**Glycolysis pathway.** Pancreatic β-cells respond to high blood glucose levels by metabolizing the extracellular glucose, triggering an increase in ATP production, which drives the closure of $K^+$ channels and the opening of $Ca^{2+}$ channels, leading to the secretion of insulin [27]. The glycolytic pathway is therefore the primary pathway modulating insulin secretion, as it initiates the steps allowing for insulin release. The pathway begins with the GLUT2 glucose transporter, which, due to an estimated high Michaelis constant ($K_m$) value, acts as a glucose sensor [28]. Specifically, the rate of glucose uptake by the GLUT2 enzyme is proportional to extracellular glucose levels, thus modulating the glycolytic flux inside of the cell and the amount of insulin released [20]. β-cells also express the glucokinase (*gk*) enzyme (called hexokinase type IV), which furthers contributes to the cells' sensitivity to glucose and acts as the rate-controlling step in GSIS [29]. The cells show low expression of the lactate dehydrogenase (*ldh*) enzyme

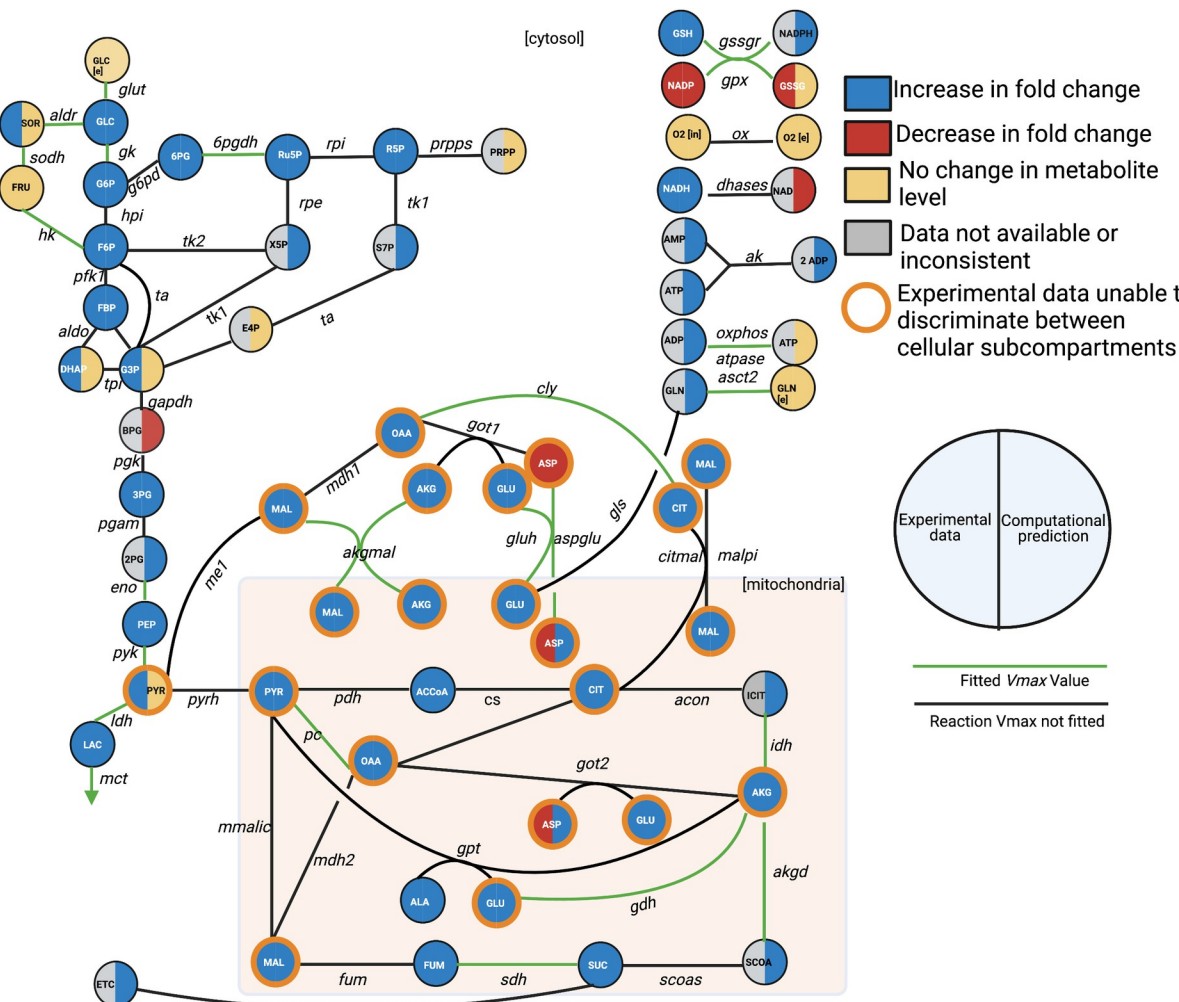

**Fig 1. Metabolic network.** The network of pancreatic beta cell central metabolism consists of 56 metabolites connected via 65 enzyme-catalyzed reactions, making up glycolysis, glutaminolysis, PPP, TCA Cycle, polyol pathway, the electron transport chain, and shuttles between the cytoplasm and mitochondria (denoted by the shaded rectangle). For clarity, some metabolites are shown as multiple nodes within a subcompartment (for example, mitochondrial AKG is shown as two nodes, though they represent the same metabolic species). As part of model training, we performed a sensitivity analysis, and the reactions whose $V_{max}$ values were significantly impactful are colored green. We compared model predictions of metabolite levels after 60 minutes of stimulation with 2.8 mM glucose, relative to metabolite levels following 60 minutes of stimulation with 16.7 mM, to qualitative shifts in metabolism as reported in the literature, provided in the supplemental material (**S1 Table**). The experimental observations are indicated by the left half of the metabolite nodes, and the model predictions are shown on the right. The model can differentiate between metabolite location (cytosol versus mitochondria), while the mass spectrometry approaches can only measure the total (pooled) amount of a metabolite; we have outlined those metabolites in orange. The full set of abbreviations and reaction equations is given in **S1 Text**.

and compensate for the need to manage NAD and NADH levels by increasing the activity of mitochondrial hydrogen shuttles, which are included in this model [30]. We also include downstream glycolytic intermediates, such as 2-phosphoglyceric acid (2PG), which have not previously been incorporated into kinetic models of pancreatic β-cell metabolism. In total, glycolysis is represented in the cytosol with 12 metabolites (2PG, 3PG, BPG, DHAP, F6P, FBP, G3P, G6P, GLC, LAC, PEP, PYR) and 13 reactions (*aldo*, *eno*, *gapdh*, *gk*, *glut*, *hpi*, *ldh*, *mct*, *pfk1*, *pgam*, *pgk*, *pyk*, and *tpi*). The glucose transport reaction (*glut*) and lactate transport reaction (*ldh*) are metabolic sources or sinks for the model, connecting the intracellular metabolites with the extracellular condition.

**Pentose-phosphate pathway.** The relationship between the PPP and insulin release is not fully characterized, as it is relatively inactive in pancreatic β-cells [31]. However, the PPP contributes to the generation of NADPH, which is believed to influence or modulate insulin secretion [32]. Furthermore, PPP metabolites have been observed to increase in abundance following glucose stimulation [33–35]. We have therefore included the pathway in the model, with seven metabolites (6PG, E4P, PRPP, R5P, Ru5P, S7P, and X5P) involved in seven metabolic reactions (*6pgdh, g6pd, prpps, rpi, ta, tk1,* and *tk2).*

**Tricarboxylic acid cycle.** TCA intermediates have been linked with GSIS, as a majority of glucose is converted into pyruvate, which is then utilized in the TCA cycle. In particular, metabolic coupling factors (MCFs) such as glutamate are known to be linked with and amplify insulin secretion [36–39]. In addition, insulin secretion is likely dependent on cofactors such as NAD or NADP, which are produced through the TCA cycle [40–42]. The TCA cycle in this model is composed of 20 metabolites across two compartments. Thirteen are found in the mitochondria (ACCOA, ALA, AKG, ASP, CIT, FUM, GLU, ICIT, MAL, OAA, PYR, SCOA, and SUC) and seven are found in the cytosol (AKG, ASP, CIT, GLN, GLU, MAL, and OAA). Twenty-five metabolic reactions carry out the import and interconversion of those metabolites (*acon, akgd, akgmal, asct2, aspglu, citmal, cly, cs, fum, gls, gluh, got1, got2, gpt, idh, malpi, mdh1, mdh2, me1, mmalic, pc, pdh, pyrh, scoas,* and *sdh).* The glutamine transport reaction (*asct2)* is a metabolic source for the model, connecting the intracellular metabolites with the extracellular condition.

**Polyol pathway.** The polyol pathway (also called the aldose reductase pathway) consists of the production of sorbitol from glucose in the aldose reductase reaction, and the subsequent conversion of sorbitol to fructose in the sorbitol dehydrogenase reaction [43]. Thus, the pathway consists of two metabolites (SOR and FRU) and three reactions (*aldr, sodh, fruT).* The polyol pathway is relatively inactive in most physiological conditions due to the aldose reductase reaction's high $K_m$ value and low affinity for glucose [44]. However, this pathway acts as a mechanism for the processing and elimination of glucose in hyperglycemic conditions, in order to protect against glucose toxicity [45, 46]. The pathway is therefore of interest within the context of β-cell stimulation with high glucose levels, though it has never been included in existing models of β-cells. Additionally, there is substantial value in studying the polyol pathway within the context of the entire metabolic network being modeled, as the PPP reduces NADPH that is oxidized by the polyol pathway, leading to a potential metabolic cycle that may impact insulin secretion [33].

**Electron transport chain.** In pancreatic β-cells, pyruvate is oxidized in the TCA cycle, generating NADH and FADH. Those reducing equivalents are then transferred through a set of electron carriers (the electron transport chain, or ETC), leading to the hyperpolarization of the mitochondrial membrane. This hyperpolarization changes the cell's ATP/ADP ratio and directly influences the release of insulin. The pathway is therefore of particular interest for understanding β-cell metabolism. The ETC in the kinetic model is composed of six metabolites (NADH, NAD, Cyt_c3, Cyt_c2, ATP, and ADP), and four reactions (*complex1, complex3, complex4, complex5).*

**Additional model metabolites and reactions.** Besides glycolysis, the PPP, TCA cycle, polyol pathway and ETC, there are other known reactions and metabolites involved in β-cell metabolism that we have included in the model. Namely, the *gssgr* reaction interconverts GSSG and GSH, *gpx* converts GSH to GSSG, *ox* transports extracellular oxygen (o2$_e$) into the cell (o2$_i$), *dhases* interconverts NAD and NADH, *ak* converts AMP and ADP into and from 2 ATP molecules, and *atpase* and *oxphos* interconvert ADP and Pi with ATP.

## 2.2 Model equations and parameters

We provide a differential-equation based model of pancreatic beta cell metabolism; as such, it calculates the simulated change in metabolite levels in time, and therefore predicts the dynamics of intracellular metabolites and the activity of metabolic reactions driving those metabolite changes. The reactions in the metabolic pathways included in the model are simulated with enzymatic reaction rate expressions, and the majority of the model reaction rate laws and overall model structure are derived from Roy and Finley's model of pancreatic ductal adenocarcinoma[22]. β-cell-specific enzyme isoforms, such as glucokinase and the glucose transporter, are taken from the published models of pancreatic β-cells from Jiang and Fridyland [20, 21]. Rate equations for the polyol pathway are taken from Cortassa et al., which simulates cardiac cell metabolism [23]. Equations for the ETC were also taken from the model published by Jiang. In total, the model contains 385 parameters, including 96 reaction velocities ($V_{max}$ values). We note that there are more reaction velocities than modeled reactions because some reactions are reversible and thus have both a forward and reverse $V_{max}$ value. To make the model specific to β-cell metabolism, we fit the $V_{max}$ values to published metabolomics data obtained from the pancreatic β-cell line INS1 832/13, as the reaction velocities often can distinguish metabolism between distinct cell types.

We calculated the left null space of the generated model's stoichiometric matrix (**S2 Table**) and therefore determined the model's conserved metabolite pools. The following pairs of metabolites were found to be conserved moieties: NADP-NADPH, GSH-GSSG, $o2_e$-$o2_i$, and mGDP-mGTP. Furthermore, assessing the model stoichiometric matrix shows that the model is indeed mass balanced, and that all sinks and sources are accounted for.

## 2.3 Initial conditions

Where available, initial conditions were taken from published studies that quantified intracellular metabolite concentrations in β-cells [21]. The initial values for the metabolites for which there was no available data were set using Latin Hypercube Sampling [47]. We specified concentration ranges based on published measurements of metabolites in other cell types and sampled 50 sets of initial conditions [48, 49]. The model was fit five times with each set of initial conditions, and we assessed the model agreement to data; the initial condition set that allowed for the best match to data was used for the subsequent fitting. This process limited the number of fitted model parameters to only the $V_{max}$ values selected with subsequent sensitivity analyses (described in section 2.5), thereby avoiding overfitting.

## 2.4 Data extraction

There have been a range of published mass spectrometry-based experiments aimed at understanding the metabolic alterations that occur in INS1 832/13 pancreatic β-cells following stimulation with extracellular glucose. We compiled those studies and extracted the metabolite fold-changes using the internet-based webplotdigitizer tool [26]. The fold-change in insulin secretion amount was similarly calculated when available.

In the study published by Spegel et al., β-cells were treated with 2.8 mM glucose for two hours and then 16.7 mM glucose for 3, 6, 10, and 15 minutes[50]. Metabolite fold-change amounts were calculated for the 16.7 mM glucose condition relative to the 2.8 mM glucose condition for 14 metabolites: 2PG, 3PG, AKG, ALA, ASP, CIT, FUM, G3P, LAC, MAL, PEP, PYR, R5P, and SUC. Insulin was also measured for the same experimental conditions. We selected the 3- and 10-minute time points as training data for use in model fitting. The 6- and 15-minute time points were withheld from model parameter estimation in order to be used for validation. Some of the measured metabolites are found in both the cytosol and mitochondria;

however, they cannot be distinguished via metabolomic analysis. Therefore, we fit the measured fold-change of the total metabolite pool. We then determined the cytosolic and mitochondrial pools separately by accounting for the relative volumes of those components, with the cytosol assumed to be three times larger than the mitochondria. Thus, we can algebraically solve for the individual cytosolic and mitochondrial concentrations.

In the work published by Malmgren et al., β-cells were treated with 2.8 mM or 16.7 mM glucose for 60 minutes [51]. Fold-changes were calculated for insulin and 14 metabolites: AKG, ALA, ASP, CIT, FUM, G3P, G6P, GLC, GLU, ICIT, LAC, MAL, PYR, and SUC. All of these measurements were used as training data.

Additionally, Spegel et al., in a 2015 paper, measured metabolite fold changes relative to the 0-minute time point, after treating with 16.7mM glucose [52]. Because those provide data relative to the initial conditions, we used the measurements to constrain the model and ensure it reaches a steady state. Data for the following metabolites were used for model training: 2PG, 3PG, AKG, ALA, CIT, F6P, FUM, G6P, ICIT, LAC, MAL, PEP, R5P, and SUC.

Our training data was therefore a combination of the Spegel et al. (3- and 10-minute) data, Malmgren (60-minute) et al data., and the Spegel et al. (6- and 15-minute) data relative to the initial time point [50–52]. Altogether, there were a total of 70 individual data points in the training set. The validation data was comprised of the 6- and 15-minute time points published by Spegel and coauthors, comprising a total of 28 distinct data points. In addition to those quantitative data points, data from eleven other published papers were used as qualitative validation of predicted fold-change direction for metabolites in the model upon treatment with above-basal glucose levels [33–35, 50–59], shown in **Fig A in S1 Supporting Information**. All experimental data used is provided in **S2 Table**.

## 2.5 Parameter estimation

In order to properly fit the kinetic model, we first performed an *a priori* parameter identifiability analysis. Specifically, we varied each model reaction rate to determine which parameter pairs may be mathematically correlated to each other and therefore structurally non-identifiable [60, 61]. We found 11 forward or reverse reaction velocities ($V_f$ or $V_r$ parameters) that were correlated to one another, and therefore defined those reaction velocities using equilibrium constants. That is, we set the reverse reaction velocity, $V_R$, to be expressed in terms of the forward reaction velocity and the equilibrium constant, $K$, so that $V_r = V_f/K$. Because of the nonlinear and dynamic nature of the model, it is possible for any two model parameters to be correlated, and therefore be structurally unidentifiable; however, for our model, the only parameters found to be correlated to each other were $V_f/V_r$ pairs for the same reaction. With this process, we therefore had 85 $V_{max}$ parameters available for model fitting and subsequent analyses.

Next, we identified the parameters that should be fit to the training data. The extended Fourier Amplitude Sensitivity Test (eFAST), a variance-based global sensitivity analysis method, was performed by simulating the same *in vitro* experimental methods used to collect the metabolomic data [62]. The eFAST method varies model inputs (the 85 $V_{max}$ parameters) two orders of magnitude above and below their baseline values in order to understand the sensitivity of the model outputs (the metabolite fold-changes). We can therefore identify the most influential model parameters to fit to experimental data, so that the model can accurately match the data without overfitting. Based on the eFAST results, we identified 33 influential model parameters by selecting the parameters that had sensitivity indices above 0.85. By only fitting the 33 most impactful parameters to the 70 training data points, we could accurately capture the observed dynamics and adequately constrain the parameter set, while avoiding overfitting.

Parameter estimation was then performed via particle swarm optimization (PSO) with the Parameter Estimation Toolbox (PESTO) in MATLAB (Mathworks, Inc.), minimizing the weighted sum of squared residuals (WSSR) [63, 64]. In order to fit the data, we computationally simulated the experimental protocols for each dataset described in Section 2.4. PSO is a stochastic global optimization tool that iteratively updates sets of randomly seeded particles (parameter sets) to converge upon a single parameter set that minimizes the WSSR. The WSSR error calculation is similar to the sum of squared residuals (SSR), but each data point is weighted by its associated standard deviation and magnitude. Weighing by the standard deviation ensures that the fitting algorithm prioritizes high-confidence data points, while weighing the error function by the data point's magnitude prevents prioritizing the high fold change measurements at the expense of the small fold changes. In the PSO protocol, each parameter is allowed to vary 100 times above and below its initial estimate, and each fitting run carries out 2,000 steps; this limits the computational cost while exploring the total parameter space. One hundred fitting runs were performed with PSO, and the best parameter sets were selected (a total of 8 best fits) based on the WSSR. These best fitted parameter values were used for all subsequent analyses and figures. This fitting approach enabled us to fit the entire training dataset (70 distinct data points, as mentioned in section 2.4) and derive consistent and high confidence parameter sets. A common feature of systems biology models is variability in the estimated parameter values due to the complexity of the model, with multiple parameter sets being equally capable of describing the system and capturing the observed data. However, performing the PSO multiple times from initial states having different initial concentrations aims to ensure the parameter estimation has reached a global minimum for the error function.

In order to justify the assumption that the kinetic parameters deemed non-impactful by the sensitivity analysis were not driving the fitted model's response, we performed additional model fits with the entire set of 85 parameters, and found the model performance did not change substantially. However, we could not ensure parameter identifiability, and could not obtain a consistent set of parameter values. Therefore, we performed model simulations using the best fitted parameters from fitting the 32 parameters shown to be influential based on the eFAST results. We use those best-fit parameter sets for all subsequent analyses.

The fitted model was therefore able to predict metabolite concentrations (in mM) and reaction fluxes (in mM/min) for each modeled species and reaction, respectively. In particular, we simulate the metabolite fold changes between high (16.7mM) and low (2.8mM) glucose conditions (Figs 1 and 2), and metabolite levels and redaction fluxes upon stimulation with high extracellular glucose alone (Figs 3 and 4).

## 2.6 Partial least squares regression

Using the trained kinetic model, we predicted the flux through each of the 65 metabolic reactions over time in 1-minute intervals. We then found the average flux through each reaction over the time period for which we had metabolomics data (3, 6, 10, 15, or 60 minutes). Thus, we predicted a single time-averaged flux value for a particular length of glucose stimulation. The predicted flux values are inputs to a regression analysis, and the corresponding fold-change amounts for insulin secretion for INS1 832/13 cells stimulated with 16.7 mM glucose relative to insulin secretion following 2.8 mM glucose from the published studies are outputs of the regression analysis.

We performed partial least squares regression (PLSR), a multivariate dimensionality reduction technique that seeks to determine a mathematical correlation between a chosen input vector with an output measurement of interest [65]. PLSR produces the components, weighted linear combinations of the inputs, that correlate with the output, and we used the SIMPLS

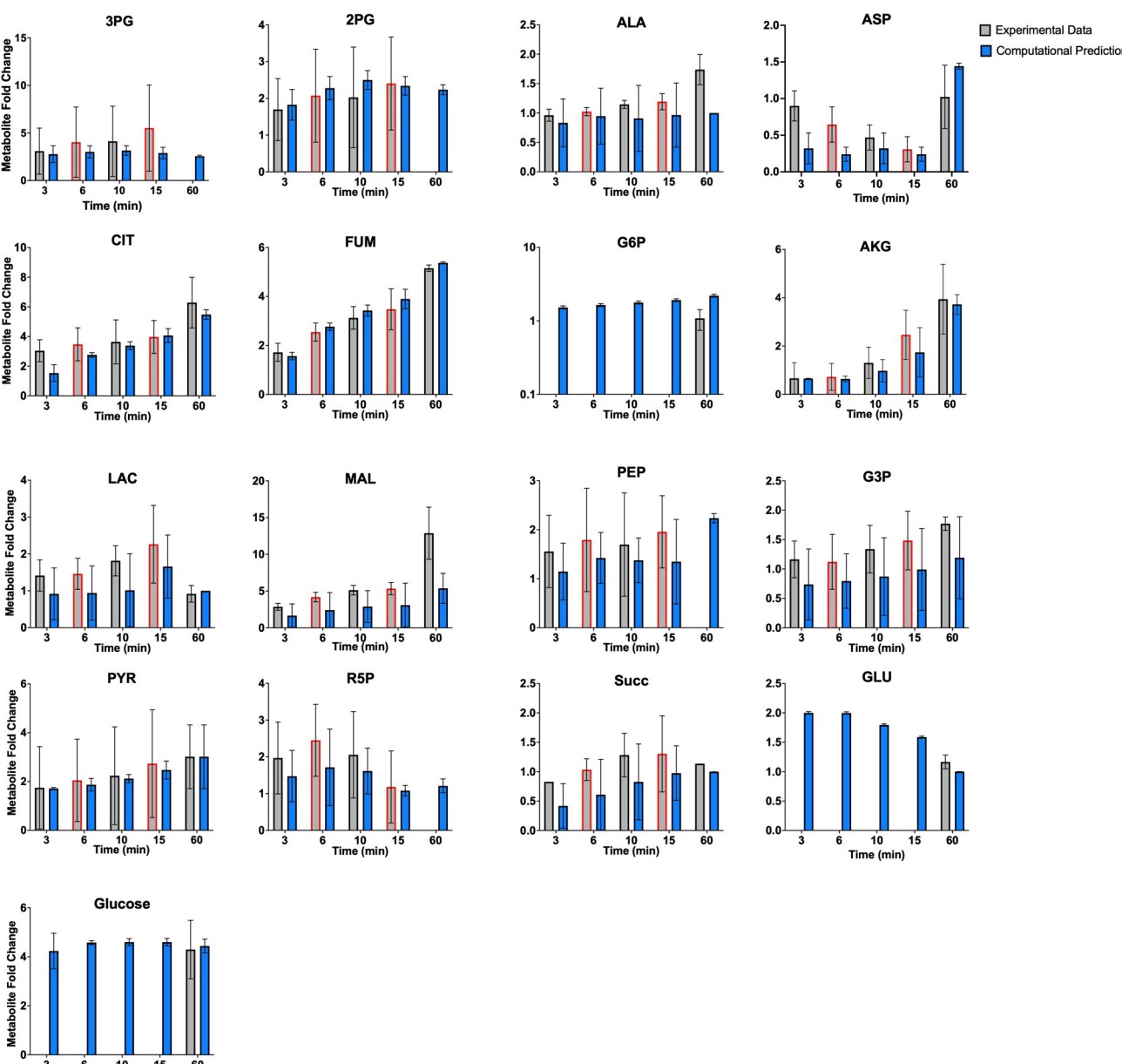

**Fig 2. Model fit to relative 16.7/2.8mM glucose experimental data.** We trained the model to mass spectrometry data published by Spegel et al. and Malmgren et al., for the 3-, 10-, and 60-minute time points (gray bars with black outline) for 17 distinct metabolites [51, 52]. Model predictions (blue bars) match experimental measurements; the error bars represent the standard deviation of model predictions across the eight best-fit parameter sets. The experimental data for the 6- and 15-minute time points (gray bars with red outline) were withheld as validation data to test the robustness of model predictions. Predicted fold-changes for metabolites found in both the cytosol and mitochondria are summed together as a total metabolite pool, in order to compare to the experimental data.

algorithm to complete this analysis [66]. The input matrix was 5 rows by 65 columns, with the columns representing the predicted average flux for each model reaction, and the rows representing the five time points of interest. The output matrix was 5 rows by 1 column, corresponding to experimentally measured insulin secretion fold-change at the five time points. By performing partial least squares regression, we specify the relationship between flow of material through the metabolic network and insulin released by the β-cell. The 6- and 10-minute

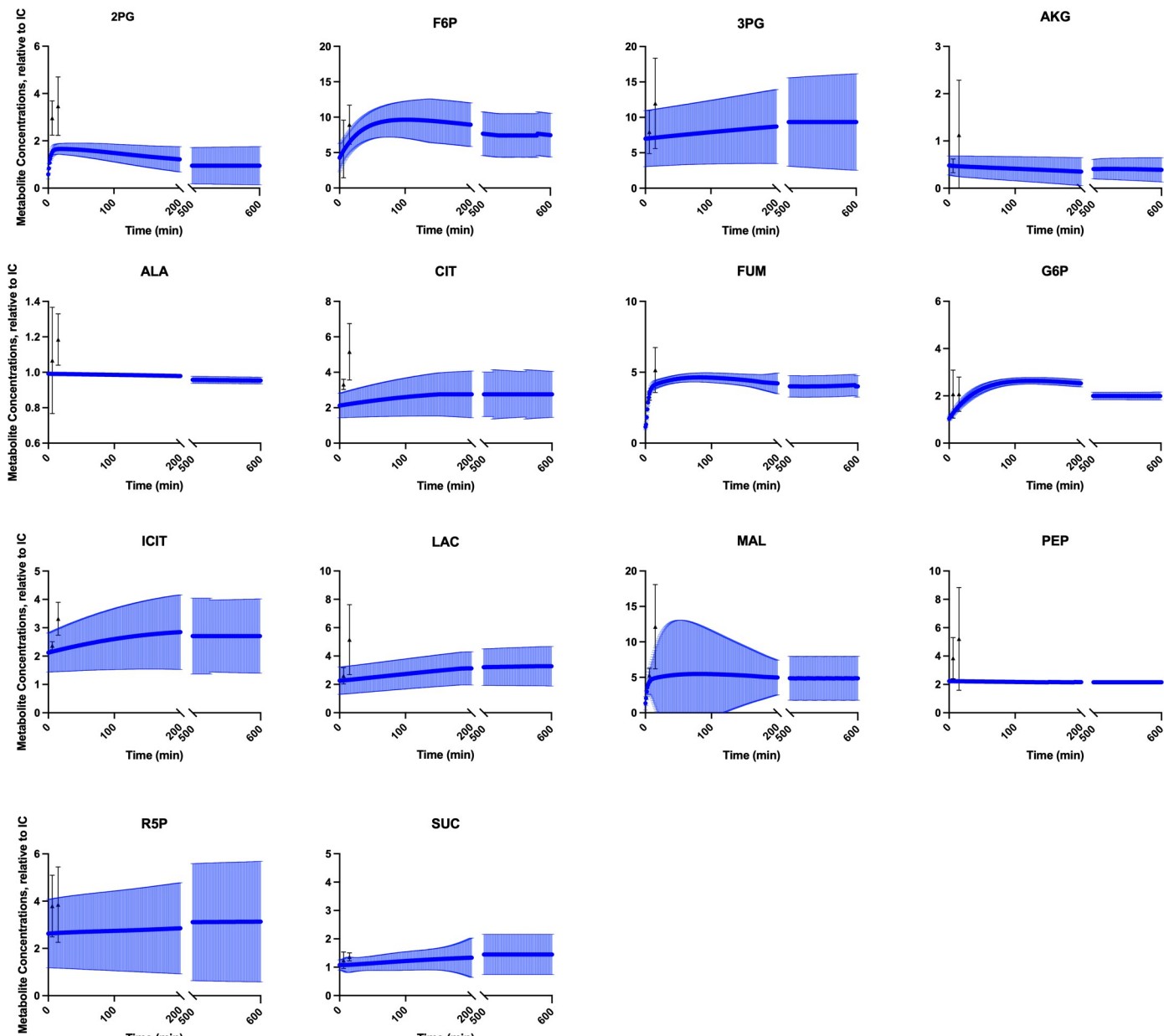

**Fig 3. Model fit to experimental data relative to initial state.** We trained the model to mass spectrometry data published by Spegel et al. for 14 metabolites (at 16.7mM glucose), relative to the initial 0-minute condition[52]. The experimental data (black triangles for measurement average, bars representing data standard deviation) are at the 6- and 15-minute timepoints. The model simulations (blue dots for average prediction, blue bars showing standard deviation) demonstrate that the model reaches a steady state condition, within 200 minutes for most metabolites. Predicted timecourses for all metabolites predicted by the model are shown in **Fig D in S1 Supporting Information**.

time points were withheld to be used as validation data, while the 3-, 15-, and 60-minute time points were used to build the PLSR model. This process was performed using each of the eight best-fit parameter sets. Thus, all predictions from PLSR analysis are showing the averaged results across the eight PLSR models. We evaluated the PLSR model fitness with the $R^2$ and $Q^2Y$ values, which range from 0 to 1. The $R^2$ value indicates model agreement to the training data, termed "goodness of fit", and the $Q^2Y$ value assesses the ability of the model to predict data not used for training, i.e., "goodness of prediction" [67].

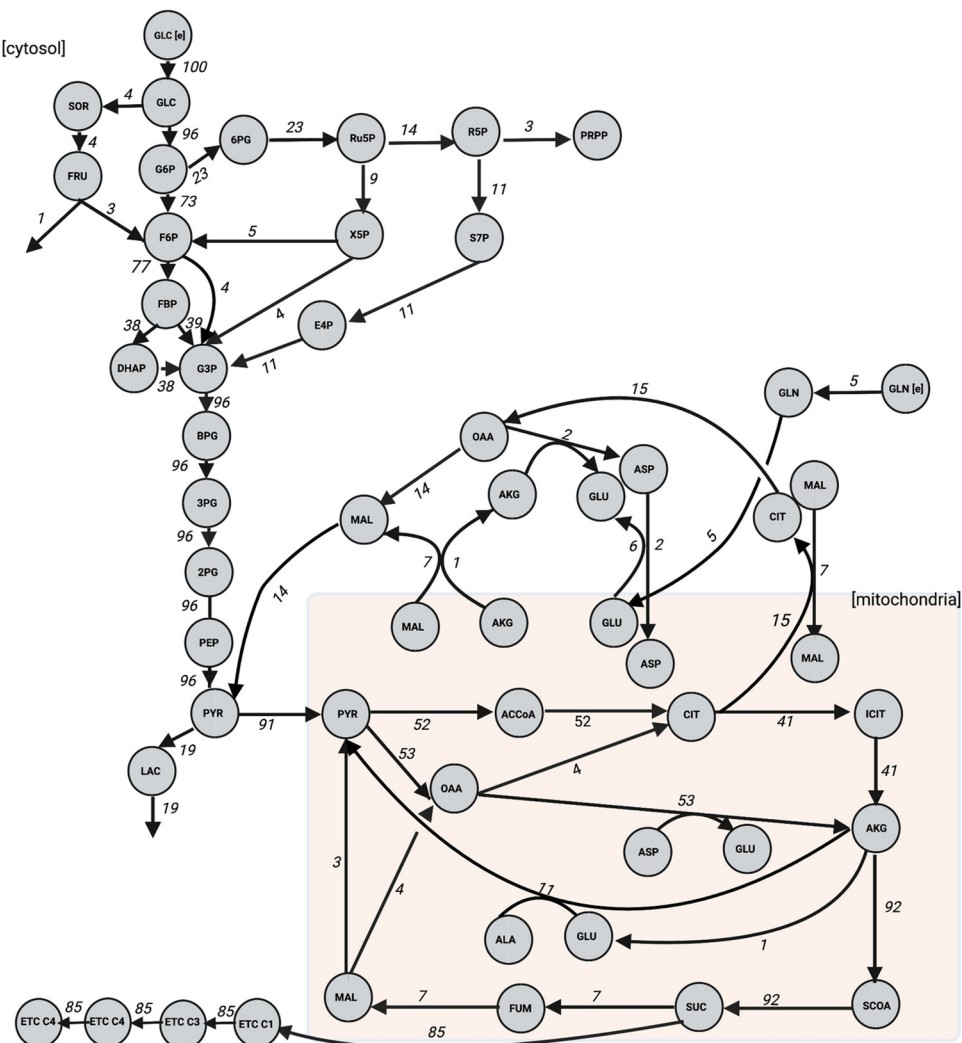

**Fig 4. Predicted steady-state reaction fluxes.** We applied the model to predict the flux through each reaction following stimulation with 16.7mM extracellular glucose. Results are presented as the flux through each model reaction when the system has reached steady state, using Parameter Set 1 in S5 Table. Flux values are shown as a percentage of the glucose transport reaction flux. We note that for clarity, some metabolites are shown as multiple nodes within a subcompartment (for example, mitochondrial AKG is shown as two nodes, though they represent the same metabolic species).

We use two quantities calculated in the PLSR analysis to gain biological insight into how intracellular metabolism influences insulin secretion. The PLSR analysis estimates the variable importance of projection (VIP) score for each reaction flux. The VIP score quantifies the contribution of each input value to the model predictions. Here, the VIP score identifies the metabolic reactions that are most strongly correlated to insulin secretion. Generally, VIP scores greater than one indicate influential inputs [68]. In addition, by assessing the PLSR model weights, we determined the effect that altering flux through the metabolic network would have on insulin secretion. For example, a negative weight indicates that increasing flux through that reaction would decrease insulin secretion. The VIP score and weights are unitless quantities.

We also performed PLSR analysis to determine how intermediate time points influence insulin secretion, to understand the time-dependent nature of the link between metabolism

and insulin at a higher resolution. In order to accomplish this goal, we again used the metabolic network's fluxome predicted by the fitted kinetic model of metabolism as an input to a PLSR model; but this time, at 1-minute intervals for a single time course. For example, the kinetic model predicted the fluxes through each reaction for the first three minutes of simulation and used this set of three predicted fluxes as inputs to a PLSR model. The fold-change in the secreted insulin calculated from the experimental data was used as the PLSR output. Because the PLSR approach requires multiple data points for both its input and output, we assumed a linear increase in secreted insulin per minute, thus giving the same number of outputs (fold-change of insulin secretion calculated per minute from the experimental data) as inputs (predicted reaction fluxes for each minute). We performed this analysis to determine the time-dependent relationships between reaction fluxes and insulin secretion over 3, 6, 10, and 15 minutes. As described above, we used the VIP scores and weights from the PLSR models to identify the most important reaction fluxes for each one-minute interval within each time course.

## 2.7 Kinetic model perturbations

**Variation of each model parameter.** The kinetic network can be used to predict the effects of metabolic perturbations on metabolite levels and the fluxes through the metabolic reactions, thereby giving a prediction of the systems-level response of the cell. Using the fitted kinetic model, we simulated metabolic perturbations, where we increased and decreased the rate of each model reaction by a factor of two and assessed the impact of that perturbation on all modeled metabolites, compared to the unperturbed baseline condition.

**Implementation of pharmacological interventions.** We selected three anti-diabetic pharmacological interventions to simulate in the model. Two perturbations are based on existing agents: metformin, which is the most common drug taken by diabetic patients but whose impact on β-cells is not fully understood, and agrimony, a medicinal plant believed to act as an antioxidant in the β-cell. Lastly, we simulated the upregulation of the adenylate kinase (*ak*) reaction, which reversibly catalyzes ATP and AMP from two ADP molecules, as that intervention was predicted to be the most beneficial for increasing insulin secretion according to the PLSR model.

Metformin primarily acts on the peripheral tissues and organs by reducing hepatic glucose production and increasing skeletal muscle glucose uptake. Together, these effects reduce hyperglycemia and effectively treat diabetes [69]. It is unclear how or if metformin affects pancreatic β-cells *in vivo*. Lamontagne et al. proposed that metformin drove "metabolic deceleration", wherein the INS1 β-cell experiences a decrease in glucose-induced insulin secretion, thereby protecting the cells from hyper-responsiveness or hyperglycemic glucotoxicity and lipotoxicity [70]. Others have shown that metformin protects against β-cell exhaustion by reducing the body's blood glucose levels [26, 71, 72]. To test this hypothesis, we decreased glucose transport into the cell by reducing the $V_{max}$ value for the *glut* reaction by 80%. We then calculated the effect on metabolites, metabolic fluxes, and insulin secretion, and compared model predictions to reported metabolomics data when possible.

Agrimony (*Agrimonia eupatoria*) is a medicinal plant used around the world to treat diabetes, especially in traditional Eastern medicine practices [26, 73]. It has been shown to affect insulin secretory activity, both in patients and in a pancreatic β-cell line *in vitro* [74]. It is believed that agrimony acts as an antioxidant in the β-cell [75]. It is well known that oxidative stress, induced by reactive oxygen species (ROS) and reactive nitrogen species (RNS), impairs β-cell activity and is a contributing factor observed in diabetes progression [76–78]. The PPP is believed to impact the oxidative stress response, as it is a one of the primary NADPH-

generating pathways. The generation of NADPH in high-glucose conditions is shown to reduce cell inflammation [26, 79]. Furthermore, an increase in PPP flux shifts the cell metabolism away from ROS-generating pathways, limiting further stress on the cells [80]. To simulate the action of agrimony, we perturbed the glucose-6-phosphate dehydrogenase (*g6pd*) reaction with a five-fold increase of its $V_{max}$ value, to simulate overexpression, as *g6pd* is the primary upstream controller of PPP activity [81, 82, 87].

Finally, based on results from PLSR modeling, we simulated the effects of perturbing the *ak* reaction. We implemented a five-fold reduction in $V_{max}$ value and assessed the effect on the metabolic network as a whole.

## 3. Results

We developed a kinetic model capable of simulating the dynamics of intracellular metabolism of the pancreatic β-cell. The model includes key metabolites and reactions involved in glycolysis, glutaminolysis, PPP, TCA cycle, and polyol pathways, with cytosolic and mitochondrial compartments. The model consists of 58 metabolites and 65 metabolic reactions and was trained to published mass spectrometry-based metabolomics data sets. The fitted model predictions were used to create a PLSR model to study the relationship between flux through the metabolic network and secreted insulin. We then assessed the impact of various metabolic perturbations on model predictions.

### 3.1 Model fitting to training and validation metabolomics data

We developed the model structure to comprise pancreatic β-cell central carbon metabolism, including isoforms and pathways unique to the β-cell. After performing a parameter identifiability analysis to exclude the forward and reverse reaction velocities that were correlated to each other, we performed a global sensitivity analysis using the eFAST method, identifying the influential model parameters to fit to experimental data (**Fig B in S1 Supporting Information**). Finally, we used published metabolomics data and performed PSO in order to find optimal parameter values. We selected the eight best-fit parameter sets based on the error between model predictions and experimental data. The estimated values for some parameters were consistent, with 10 of the 32 parameters varying less than 10% across the eight best fits (**Fig C in S1 Supporting Information**). Interestingly, the $V_{max}$ values for several reactions exhibit bimodality across the eight fitted parameter sets, showing multiple regimes that are equally capable of explaining the data.

The trained model was able to closely match the quantitative fold-change values measured experimentally for 17 metabolites used for parameter estimation (**Fig 2**). Furthermore, the model matched the fold-change data for time points that had been withheld as validation data, pointing to the ability to successfully predict data not used for model training. This further establishes the model's ability to match experimental data. We also compared model predictions to the qualitative direction of the change in metabolite levels upon stimulation with high glucose, based on literature review. These observed changes are indicated in the coloring of the nodes in **Fig 1** and the squares in **Fig A in S1 Supporting Information**. Additionally, we performed a statistical analysis on the model fits, comparing each model prediction with its corresponding experimental data with a *t*-test. Only six of the 99 *t*-tests (3-minute CIT and 60-minute MAL for the high-glucose relative to low-glucose comparisons, and the 6- and 15-minute 2PG, 15-minute ALA, and 15-minute CIT for the metabolite comparisons relative to their initial conditions) found a statistically significant difference between the predictions and data. This analysis indicates that the model predictions match data well. The results from this statistical analysis are given in the supplement as **S3 Table**.

Besides the 17 unique metabolites used for model fitting, the model also predicts the fold-changes of 31 distinct metabolites. Most metabolites (88%) qualitatively match the experimental fold-change direction, even metabolites that were not used in training.

The model is also capable of capturing the dynamics of biomarkers known to be important in GSIS. For example, the models show a 6.2-fold increase in the NADPH/NADP ratio after stimulation with high glucose for six minutes, as compared to an 8-fold increase reported by Spegel 2013. This ratio is considered a metabolic coupling factor for insulin and is therefore an important metric for confidence in the model [83–85]. This demonstrates that the model predictions are closely aligned with data not used for model training.

In addition to confirming that the model is capable of recreating observed metabolite levels for multiple glucose stimulation levels, we also wanted to ensure that could capture a steady state condition. We therefore simulated the model with 16.7mM extracellular glucose for 72 hours to demonstrate that the predicted metabolites do not steadily increase, but approach an equilibrium condition. We provide the first 600 minutes in **Fig 3** and show that the model metabolites evolve towards a steady state while agreeing well with the experimental data (shown in black).

We also applied the model to predict the flux distributions of the cell (**Fig 4**), corresponding to the steady state conditions attained by the metabolites in **Fig 3**. Because the metabolite levels are at steady state, the sum of fluxes into each node of the network equals the fluxes leaving the node, and the total flux into the system will equal the sum of fluxes out of the network (**Fig 4**). The model predicts fluxes in units of mM/min, and we present the flux values relative to glucose import into the cell, thus assigning the glucose transport reaction flux a value of 100 and allowing all other reaction fluxes to be seen as a percentage of that import flux. This gives insight into how glucose is utilized in the metabolic network.

Though there is a paucity of experimental data with which to compare the flux predictions, Cline et al., Shi et al., and Berman et al. have quantitatively measured the activity of various metabolic reactions in INS1 cells [86–88]. Although we cannot make a close, direct comparison given differences in incubation and treatment of the cells, the model predicted fluxes show a close correspondence with the experimentally measured reaction fluxes (**Fig E in S1 Supporting Information**). Four out of the five predicted fluxes are within the error of the experimental data, which were not used for model training. Therefore, with both predictions of metabolite concentrations and reaction fluxes, the model can provide a systems-level understanding of the effect of glucose stimulation.

## 3.2 Partial least squares regression modeling

We developed a PLSR model correlating the reaction fluxes predicted by our calibrated model to reported insulin secretion. For each of the eight best-fit parameter sets, we predicted the flux through each reaction for the time period used in the experimental studies. We then performed PLSR analysis with the predicted fluxes as inputs and the measured fold-change in insulin secretion as outputs. Because each parameter set produced a distinct set of reaction fluxes, we generated and analyzed eight separate PLSR models. We found that PLSR models with three PLSR components best represented the data, capturing the majority of the variation in the outputs (**Fig 5A**).

The PLSR models agreed well with the experimental data (**Fig 5B**), both for the data used in training and data withheld for validation. The average $R^2$ value across the models was 0.95, and the $Q^2Y$ was 0.74. We note that the low $Q^2Y$ value is likely due to the low number of time points used to develop the model, as the $Q^2Y$ performance metric assesses how the model would perform if trained on a subset of the available data and asked to predict the withheld

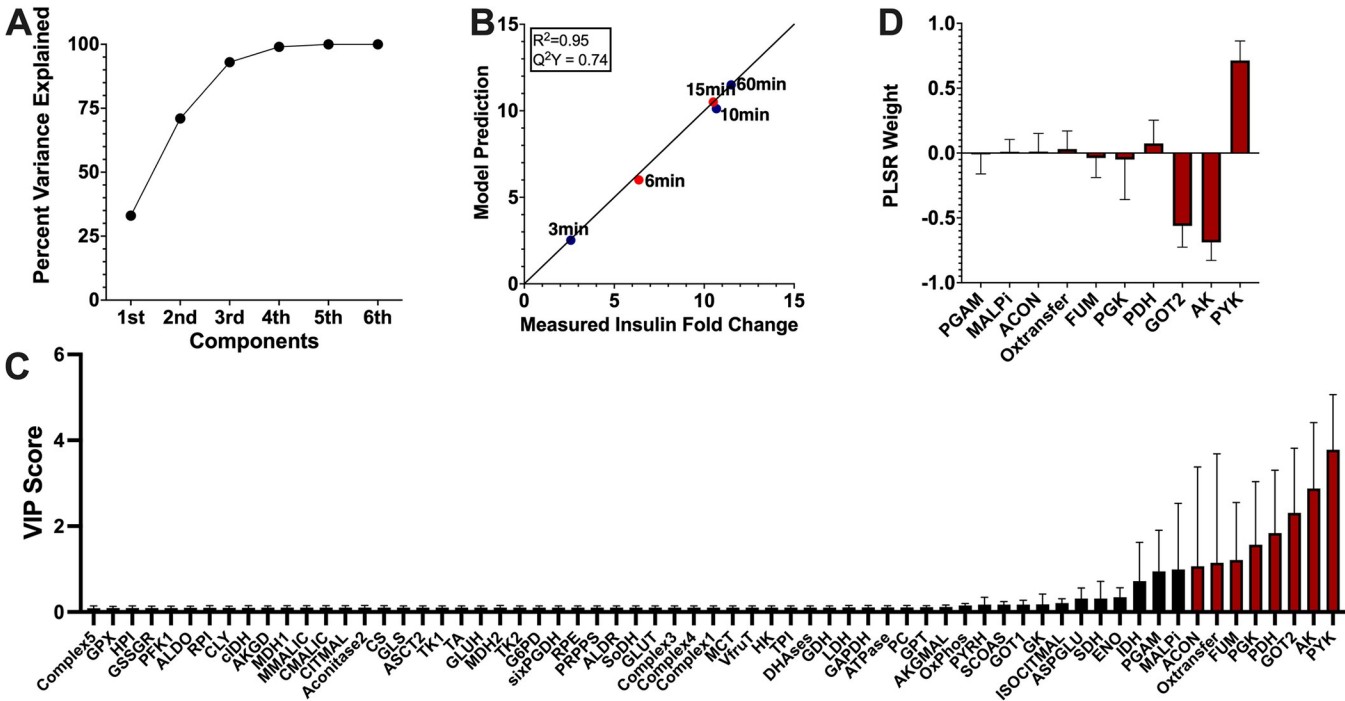

**Fig 5. Partial least squares regression analysis.** The PLSR model correlates the predicted flux through each reaction with measured insulin secretion amount at each time point of interest. Results shown are the average prediction across the PLSR model from each of the eight best-fit parameter sets. **(A)** Three PLSR components were used, as they collectively accounted for most of the variance. **(B)** Model predictions agreed with reported insulin amount, both for time points used for PLSR model building (3, 10, and 60 minutes, blue circles), and for those held out as validation (6 and 15 minutes, orange circles). Results are shown as fold-change in insulin secreted for 2.8mM glucose compared to 16.7mM. The PLSR models had an average $R^2$ value of 0.95 and $Q^2Y$ value of 0.74. **(C)** We assessed the VIP scores for each metabolic reaction flux, shown in increasing order. Reactions with VIP scores greater than one are shown with red bars. **(D)** We analyzed the PLSR model weights associated with each reaction flux that were determined to be impactful. The weight shows how a change in flux value will affect insulin secretion. A positive weight indicates that increasing the flux value will increase insulin secretion.

data (leave-one-out cross validation). For a PLSR model with a small number of rows in the input matrix, leaving any data point out can substantially change the model's predictive power. Thus, the somewhat low $Q^2Y$ value is to be expected. Overall, this data-driven regression analysis approach reliably predicts the relationship between the flow of material through the network and the insulin produced.

We used the PLSR models to estimate the VIP scores, identifying the most important reaction fluxes that drive insulin secretion (**Fig 5C**). The reactions with VIP scores greater than one are colored red, and the predictions are consistent across the eight models.

Interestingly, several reaction fluxes with a VIP score above that threshold are either involved in glycolysis (*pyk* and *pgam)*, the synthesis of energy (*oxtransfer* and *ak*) or the conversion of TCA cycle metabolites (*got2, fum, pdh, malpi,* and *acon)*. The PLSR model weights for reactions with VIP scores above 1 (**Fig 5D**) provide information on whether a reaction is positively or negatively correlated to insulin secretion, and the strength of that correlation. *Ak, got2, pgk, fum,* and *acon* are negatively correlated in the model, suggesting that knocking down the reaction will drive an increase in insulin secretion. The pyk, *pdh*, *oxtransfer*, *acon*, and *malpi* reactions were positively correlated with released insulin.

We also developed PLSR models at 1-minute time intervals for each short time course individually (3, 6, 10, and 15-minutes). We again assessed the importance of each metabolic reaction on insulin produced per minute in a given time period. These results are shown in (**Fig 6**). Though the reactions with high VIP scores (pyk, *ak*, *got2*, *fum*, *oxtransfer*, and *acon*) are

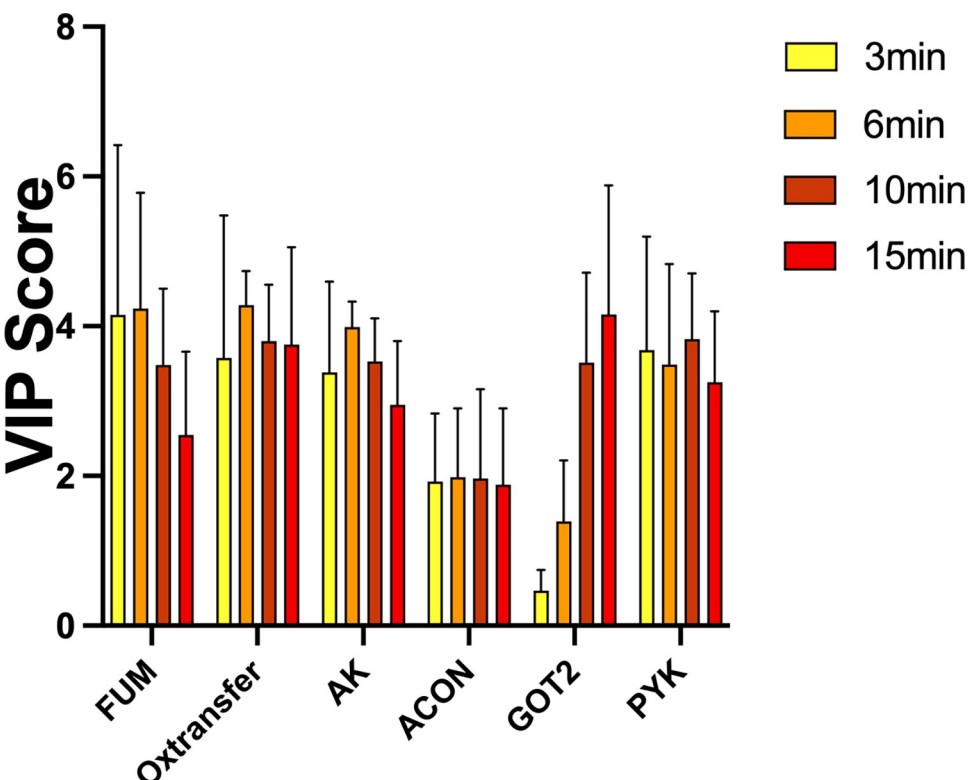

**Fig 6. Short-term time course PLSR models.** PLSR models were generated for each shorter time course of interest (3, 6, 10, and 15 minutes), correlating the average flux through each reaction with the insulin produced. The predicted VIP scores are shown for each time course.

consistent between the total treatment time (**Fig 5C**) and the short time courses, the oxygen transfer reaction emerges as important in the short term. Interestingly, though the same five reactions are consistent in the short-term time, the VIP scores differ across the different time points. *Got2* grows increasingly more impactful over time, while *fum* is most influential during the initial time points. Overall, the kinetic and PLSR models allow us to predict targets for modulating intracellular metabolism and insulin secretion.

### 3.3 Effects of varying each model $V_{max}$ value

By combining the kinetic and PLSR models, we linked intracellular metabolism and insulin secretion. Using the integrated modeling framework, we predict how perturbing metabolic reactions affects insulin secretion and the whole metabolic network. We knocked down and increased the $V_{max}$ value of each metabolic reaction by a factor of two and assessed the impact on each metabolite and on insulin secretion. The predicted fold-changes in the metabolite levels compared to the baseline model with no perturbation are shown in **Fig 7**, for all parameter values that elicit a change in any prediction.

As expected, increasing a $V_{max}$ value led to the opposite effect as decreasing the value for most reactions, but the amount by which those opposing perturbations affect cellular metabolism is not equal for every reaction velocity. For example, increasing the pyruvate dehydrogenase (*pdh*) reaction causes substantial decreases in TCA cycle intermediates, but decreasing the rate does not lead to a comparable change in metabolite amounts. Similar trends can be seen with changing pyruvate carboxylase (*pc*), the cytosolic malic enzyme (*cmalic*), and

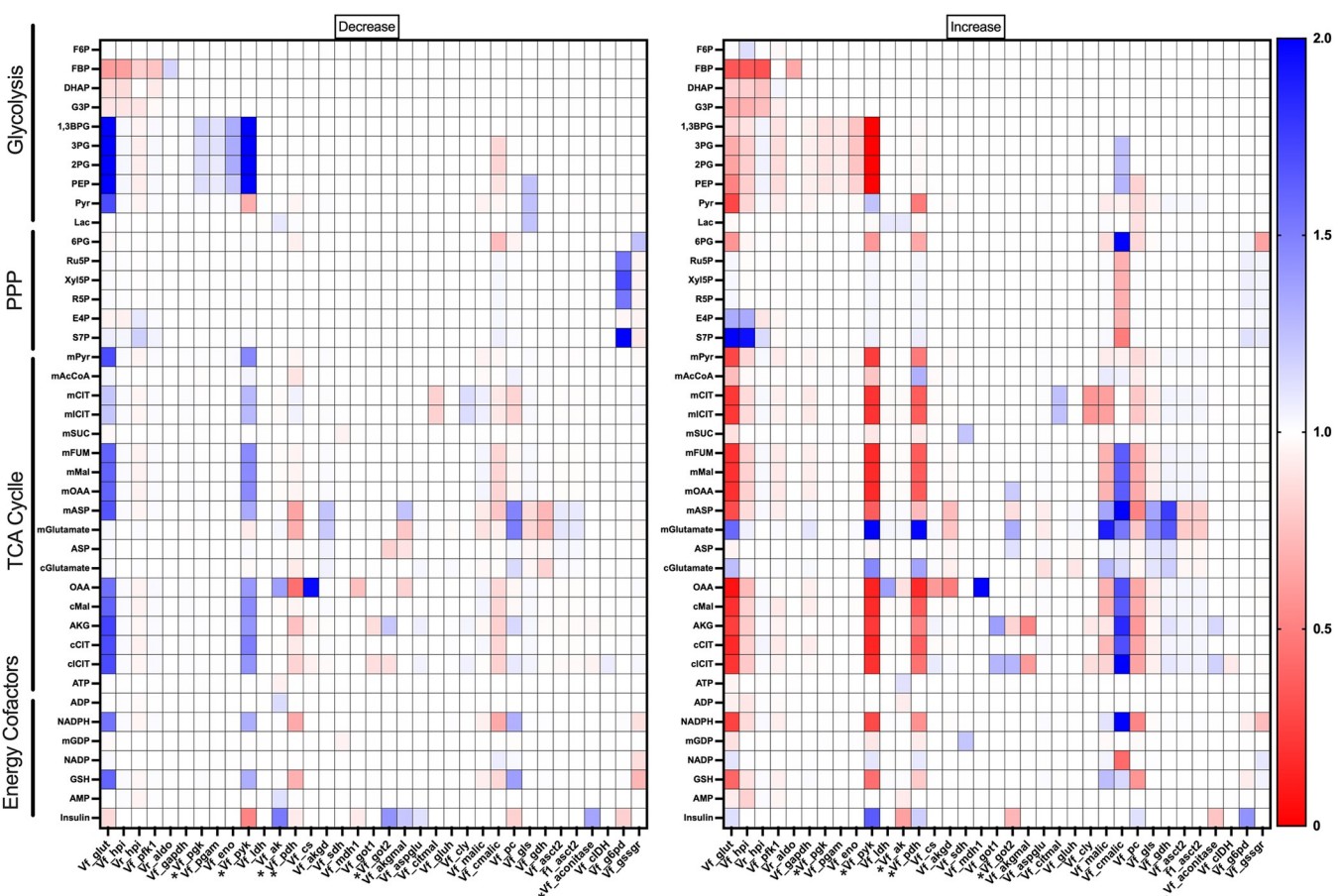

**Fig 7. Effects of metabolic perturbations.** We decreased **(left)** and increased **(right)** each reaction $V_{max}$ value (*y*-axis) by a factor of two and assessed the impact on all metabolites and insulin (*x*-axis). Metabolites and parameters that did not change or cause any changes, respectively, were excluded from the figure for better readability. The color bar indicates the effect of the perturbation relative to the base model with no perturbation. Parameter values which were found to be influential in the PLSR analysis are marked with a star.

glucose-6-phosphate dehydrogenase (*g6pd*), where the direction of the perturbation substantially and differentially impacts the metabolite levels in one direction more than the other. Perturbing some metabolic reactions is only impactful in one direction; for example, increasing flux through the glucose transport reaction will lead to an increase in S7P, but decreasing that reaction will have no impact on the metabolite levels.

Considering all of the metabolites in the model, we predict that downstream glycolytic metabolites (1,3-BPG, 2PG, 3PG, and PEP) are particularly susceptible to perturbations in the rest of the network, irrespective of the direction of that perturbation. Similarly, mitochondrial aspartate and glutamate are sensitive to changes in the network, likely due to their ubiquity in the metabolic processes. In addition, perturbing the *glut*, *pdh*, *pyk*, and *pc* reactions elicit widespread changes in metabolite levels, suggesting that the reactions are primary control points in the network that could cause a drastic shift in metabolism if targeted. Interestingly, perturbing the glucose transporter (*glut*) impacts the levels of certain TCA cycle metabolites (mitochondrial fumarate, malate, oxaloacetate, and aspartate; cytosolic oxaloacetate, malate, α-ketoglutarate, and citrate) whether the enzyme $V_{max}$ value is increased or decreased. This indicates that the influx of glucose into the cell exert tight control over metabolic changes, as would be expected.

The regression analysis gives complementary information to the kinetic modeling, as it shows the metabolic reactions that are related to insulin. The VIP scores show which reaction fluxes correlate to insulin secretion, while the weights show whether those fluxes are positively or negatively correlated. We increased and decreased each metabolic reaction flux by a factor of two, and assessed the resulting predicted change in insulin secretion. These results are shown in the rightmost row of **Fig 7**. As indicated by the VIP scores and weights from the PLSR model (**Fig 5**), increasing the *pyk* and *pdh* reaction fluxes are predicted to increase insulin secretion. Increasing the *ak* or *got2* reaction fluxes leads to a decrease in insulin produced. Interestingly, reactions with lower VIP scores are also predicted to affect insulin secretion, including the *glut*, *g6pd*, and *pc*, since adjusting the $V_{max}$ value for these reactions affects insulin secretion.

## 3.4 Effects of metformin on pancreatic β-cell metabolism

Metformin was initially discovered as an antimalarial agent, but has become the leading diabetes drug in use because of its ability to lower blood glucose levels in the body [89]. Metformin primarily acts on the peripheral tissues and organs, reducing hepatic glucose production and increasing skeletal muscle glucose uptake. Though it is unclear how or if the drug affects pancreatic β-cells, one proposed hypothesis is that it reduces the cells' uptake of glucose, thereby acting as a protective mechanism to avoid overactivity and cellular exhaustion [26, 70, 72]. We tested the effects of metformin with our integrated modeling framework by decreasing the $V_{max}$ value of the *glut* reaction by 80%. We analyzed the impact on the predicted insulin amount and the metabolic network (i.e., metabolites and reaction fluxes).

The PLSR model ranked the flux through the glucose transporter as a relatively uninfluential reaction, with an average VIP score of 0.3. Thus, it is not unexpected that decreasing the $V_{max}$ of the *glut* reaction does not change the predicted insulin secretion. This is consistent with the field's consensus that the availability of glucose, and not its transport rate, is believed to be the driving factor modulating insulin secretion. This is because the glucose transporter has a high $K_m$ value, which causes its observed "glucose sensing" ability [1, 90].

We then assessed the impact of the simulated perturbation of the *glut* reaction on the kinetic model (**Fig 8**). The predicted metabolite fold-change are given in **S4 Table**. As expected, we see a decrease in intracellular glucose levels. Similarly, upstream glycolytic metabolites are predicted to decrease, as do most PPP and TCA cycle intermediates. Flux through the reactions involving those metabolite levels is also substantially decreased, compared to the unperturbed system. Due to the simulated metformin perturbation, the *fum*, *akgmal*, and *rpi* reactions proceed in the opposite direction compared to the base model. Interestingly, the levels of downstream glycolytic metabolites (1,3-BPG, 3PG, 2PG, and PEP) are predicted to increase due to the simulated perturbation. This indicates that metformin leads to an accumulation or pooling of those metabolites. This accumulation is further confirmed by the prediction that flux through the glycolytic reactions involving these species (*pgk*, *pgam*, *eno*, and *pyk*) decreases.

The simulated perturbation of glucose transport also affected some nucleotides, as we see a reduction in cellular ADP and AMP levels. However, ATP, NAD, NADH, NADP, and NADPH levels are mostly unchanged. The predicted decrease in ADP and AMP levels is driven by a decrease in the *ak*, *ox*, and *dhases* reactions, and an increase in the *atpase* reaction. Thus, the model predicts that the levels of the high-energy metabolite are robust to perturbations in glucose uptake.

## 3.5 Predicted effects of agrimony on pancreatic β-cell metabolism

*Agrimonia eupatoria* (also called church steeples, in the Rosaceae family) is a traditional medicinal herb used to treat diabetes, as it has been shown to promote insulin secretion [74].

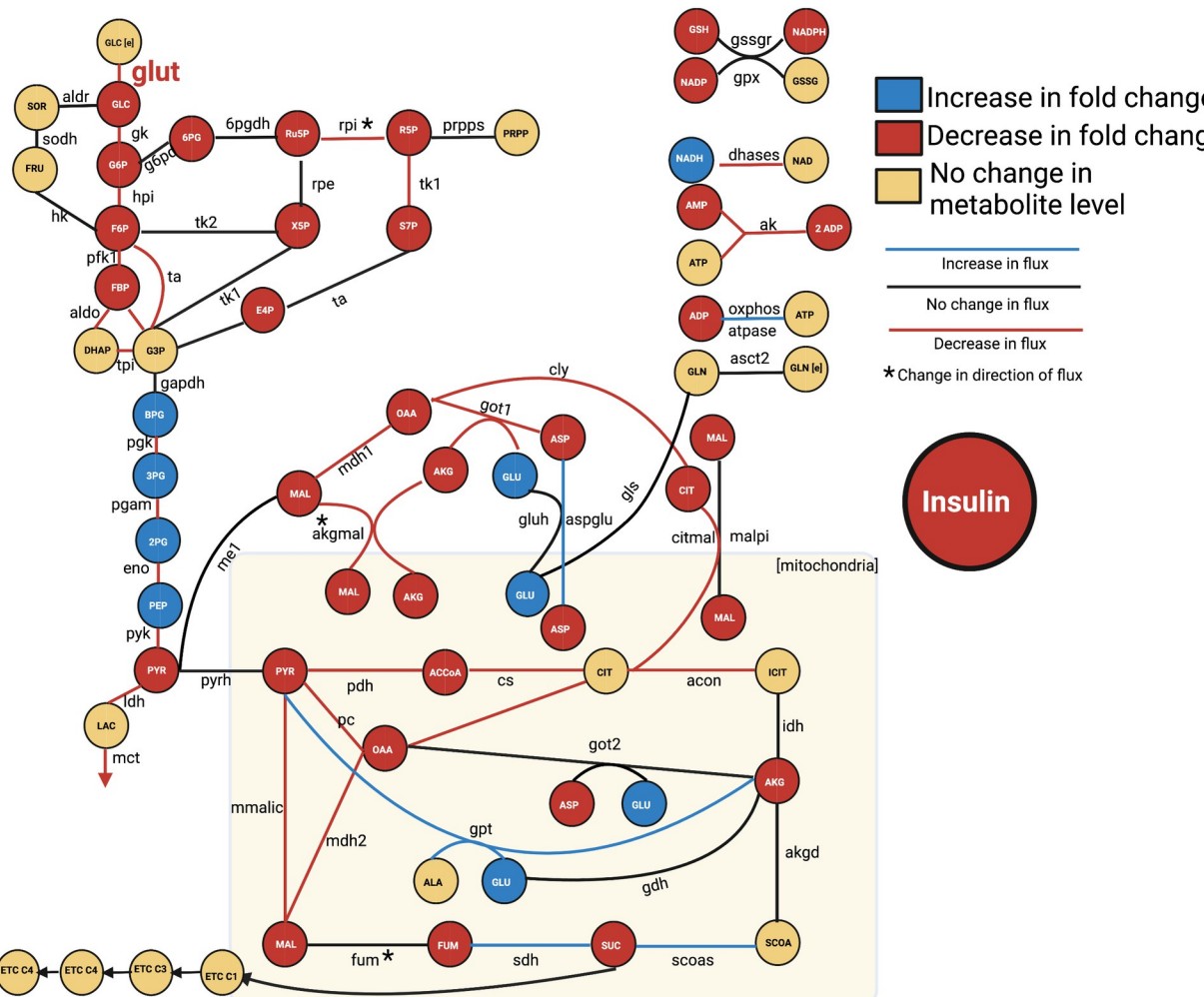

**Fig 8. Effect of metformin treatment.** We implemented metformin as an 80% knockdown of the glucose transport (*glut2*) reaction and assessed the effect on the network, comparing metabolite levels, reaction fluxes, and insulin secretion to the unperturbed condition.

A common proposed mechanism by which agrimony induces increased insulin release is through antioxidant activities. It is well understood that diabetes and many other chronic illnesses are mediated through chronic inflammation, often driven by reactive oxygen or nitrogen species, which are affected by antioxidants [91–94]. It has been shown that the PPP serves to reduce inflammatory species, as the pathway drives the production of NADPH in the cell, which exerts a protective and anti-inflammatory influence on the β-cell [40–42, 79]. To simulate the action of agrimony, we perturbed the glucose-6-phosphate dehydrogenase (*g6pd*) reaction by increasing its flux, to simulate overexpression, as *g6pd* is the primary upstream controller of PPP activity.

The PLSR model predicted relatively minor increases in the insulin secretion of the cell (**Fig 9**), reported by a low VIP score. Though the metabolites and metabolic fluxes in the PPP are predicted to markedly increase compared to the baseline model condition, the perturbation caused relatively few other changes in the kinetic metabolic network: the model predicts no change in the metabolite levels or metabolic reaction fluxes in glycolysis, the TCA cycle, or the polyol pathway (**Fig 9** and **S4 Table**).

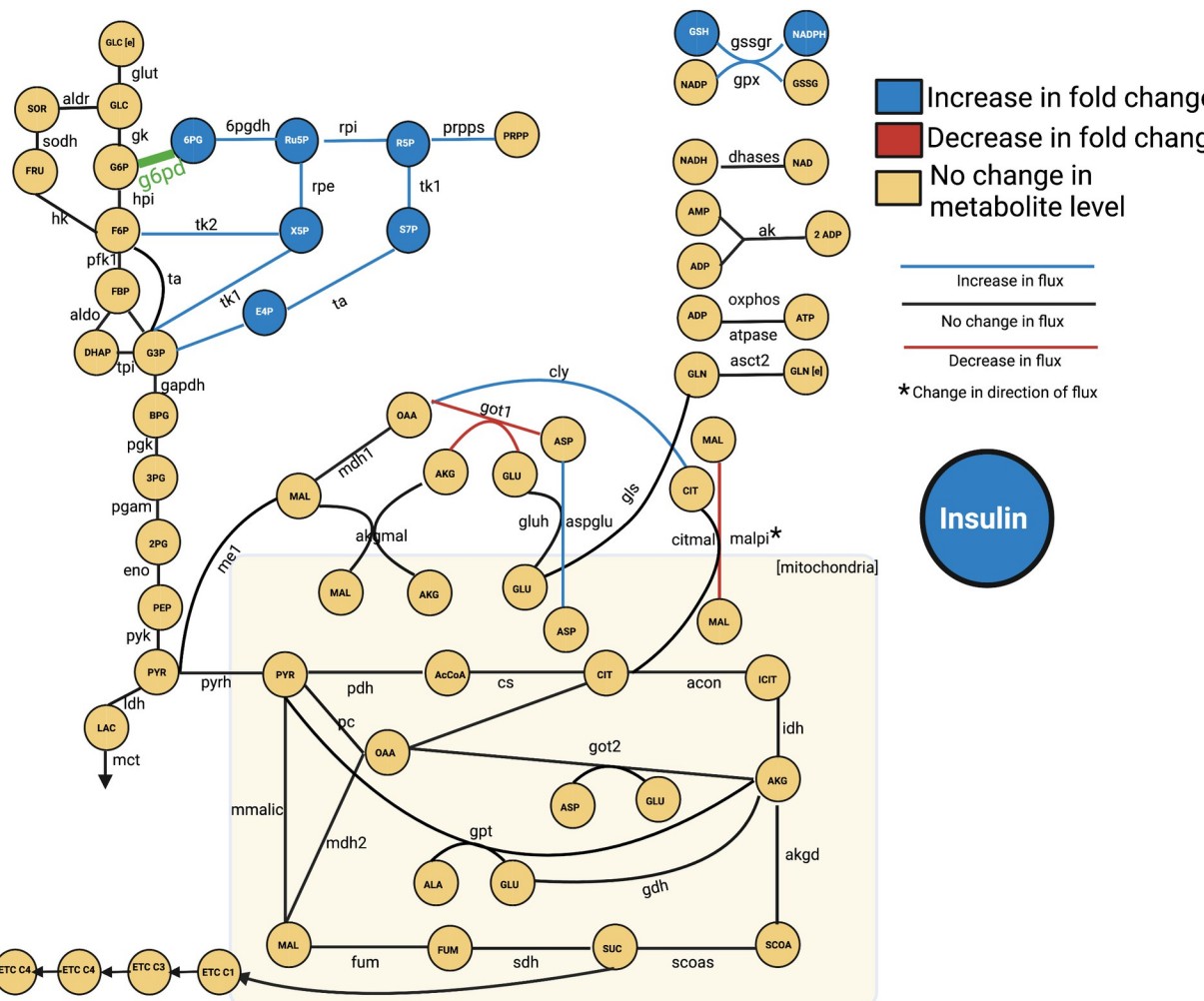

**Fig 9. Effect of agrimony treatment.** We implemented agrimony as a 5-fold increase in the *g6pd* reaction, and assessed the effect on the network, comparing metabolite levels, reaction fluxes, and insulin secretion to the unperturbed condition.

Additionally, the model predicts that increasing flux through the *g6pd* reaction does not substantially affect the levels of energy precursors, ADP, ATP, NADP, NADH, and NAD. NADPH is the only energy precursor predicted to increase in response to increasing flux through the *g6pd* reaction. This indicates that effect of *g6pd* perturbation is highly specific. Glutathione (GSH) levels in the cell are also predicted to be affected by the simulated targeting of *g6pd*, as NADPH is used to produced GSH in the glutathione reductase (*gssgr*) reaction.

### 3.6 Predicted effects of perturbing the *ak* reaction

The VIP scores and weights calculated in the PLSR analysis indicate the most impactful reaction fluxes involved in pancreatic β-cell metabolism and suggest the direction in which changing the associated $V_{\max}$ values would shift insulin secretion. Of the reactions with VIP scores greater than one, the *ak* reaction, which converts ATP and AMP into two molecules of ADP, was predicted to be among the most impactful reactions correlated with insulin. The *ak* reaction is the primary mechanism by which cells maintain adenine nucleotide homeostasis, and it affects the AMP-activated protein kinase (AMPK) signaling cascade. The role of AMPK in

insulin release is disputed, as it has been described both as a positive and a negative regulator of insulin secretion [95]. However, most published work affirms a link between *ak* and the ATP-mediated potassium channels, making the reaction of particular interest in β-cells. Thus, we reduced the $V_{max}$ value for the *ak* reaction and assessed the effect on the entire metabolic network.

In our PLSR model, the *ak* reaction flux is strongly negatively correlated to the secretion of insulin, as it has a high VIP score and negative weight. Our integrated modeling approach predicts that a five-fold decrease in the $V_{max}$ value for the *ak* reaction leads to a 1.38-fold increase in insulin secretion compared to the baseline model. However, decreasing its $V_{max}$ is not predicted to significantly affect metabolites in the kinetic network (**Fig F in S1 Supporting Information** and **S4 Table**). This may identify *ak* as a candidate treatment target for diabetes, as it could be used to increase insulin secretion, without disrupting the homeostasis of the β-cell.

## 4. Discussion

### 4.1 Utility of our predictive modeling framework

Computational modeling is a tool by which we can synthesize disparate information and data to generate novel predictions. In particular, modeling allows us to take a systems-level view of how individual parts of a metabolic network (i.e., reactions and metabolites) work together to generate observed behavior. Here, we have developed a predictive kinetic model that is able to capture the dynamics of metabolism in pancreatic β-cells. The model consists of glycolysis, glutaminolysis, the PPP, the TCA cycle, polyol pathway, and electron transport chain, building upon previously published modeling efforts. The model has been trained to and validated with published qualitative and quantitative metabolomics data from the INS1 832/13 cell line collected *in vitro*. The calibrated kinetic model predicts metabolite concentrations and reaction fluxes. It is important to note that the computational model can differentiate between the levels of metabolites that are found in both the cytosol and mitochondria, whereas the mass spectrometry pipeline pools them together and cannot easily discriminate between metabolites in different cellular sub-compartments. This is a further benefit of the kinetic modeling approach, as it can investigate the proportion of a metabolite pool in a particular compartment.

We paired the kinetic model with regression analysis. Integrating a kinetic model with a PLSR model allows us to further analyze systems-level dynamics of the central carbon metabolic network in β-cells and relate the predicted metabolite levels and reaction fluxes to a cellular-level response (insulin secretion). Though pairing kinetic modeling with data-driven modeling is a somewhat underutilized approach, it can be used to extend the predictive capabilities of kinetic modeling, thereby gaining novel insights.

The influential reactions predicted by combining kinetic and PLSR models agree with experimental observations. Our approach predicted that reactions involved in energy synthesis and TCA cycle activity strongly contribute to insulin secretion. Both of those cellular processes have been previously implicated with insulin production [26, 96, 97]. The *ak* reaction was predicted to be among the most impactful metabolic reaction affecting insulin secretion and to have a negative correlation with insulin release. It has previously been shown that *ak* is a negative regulator of insulin secretion; for example, knocking out *ak* substantially affected the stimulatory activity of the $K_{ATP}$ channels that drive insulin secretion [98–100]. TCA cycle reactions (namely, *got2*, *fum*, and *acon*) also emerged as impactful. Both mitochondrial and cytosolic TCA cycle signaling has been implicated with insulin secretion. *Got2* has not been studied in depth in this context, but, interestingly, that reaction is substantially reduced in the β-cells of neonatal mice. This potentially suggests a link between *got2* and insulin secretion, as neonatal mice fail to show proper glucose responsiveness [101, 102]. Fumarase was predicted to be

negatively related to insulin secretion; mice with fumarase-deficient β-cells developed diabetes, suggesting that proper regulation of the β-cell GSIS system is dependent on proper function of the *fum* reaction [11, 103]. Similarly, *acon* was correlated to insulin secretion. Though the mechanisms are unclear, a possible explanation is that *acon* activity is influenced by nitric oxide (NO) damage to the β-cell, which is also heavily involved in the insulin secretory activity of β-cells [104]. We also predict the importance of the pyruvate kinase enzyme in glycolysis. This enzyme has a key regulatory role in GSIS, as it is believed to contribute to the regulation of the signal strength of insulin secretion and the ATP/ADP ratio in the cell. [105, 106].

A goal of our modeling work is to establish a quantitative framework that can be used to identify novel mechanisms to treat type 2 diabetes, which involves β-cell dysfunction [107, 108]. As a step towards this goal, we perturbed each model reaction and explored in detail the effect of pharmacologic interventions, both on metabolite levels and fluxes and on insulin secretion. We thus used the model for hypothesis generation and testing. Additionally, our modeling framework can identify the time-dependent nature of insulin secretion, as shown by comparing the overall PLSR model (generated with all five time points) with the short-term PLSR models generated. We predict that the oxygen transfer reaction is important only in the short time course. Because pancreatic β-cells utilize mitochondrial metabolism, the cells demonstrate a high oxygen consumption rate. It is notable that the metabolic reaction emerges as impactful, as this highlights the time-dependent and dynamic nature of β-cell activity and suggests it may be worthwhile to investigate the impact of perturbing the oxygen transport reactions *in vivo*.

Excitingly, our model predictions of the effects of metabolic perturbations induced by pharmacologic agents agree with literature evidence. Hasan et al. suppressed pyruvate carboxylase *(pc)* reaction activity in INS1 cells and observed that lactate and pyruvate levels increased, and that malate and citrate levels decreased; our kinetic model agrees with those experimental measurements, as seen in **Fig 6** [109]. Guay et al. showed that knockdown of the malic enzyme caused a decrease in glucose oxidation (the steps converting glucose to pyruvate). Our model also predicts a decrease in glycolytic intermediates (1,3BPG, 2PG, 3PG, and PEP) upon suppression of malic enzyme [110]. We do find that though MacDonald et al. show the *idh* knockdown changing NADPH/NADP ratio, our model does not predict a change in metabolite levels when *idh* is decreased [111]. Overall, our model predictions are well supported by experimental results, lending great confidence to the model.

Lamontagne et al. treated INS1 cells with metformin at varying extracellular glucose levels [70]. They showed that metformin caused a decrease in GSIS at intermediate glucose conditions, which supports the PLSR model's prediction, as the glucose transporter has a negative, albeit small, weight, indicating a negative correlation with insulin. Lamontagne and coworkers also measured metabolite levels following metformin treatment, proposing that the treatment increased cellular glutamate levels, did not affect the concentrations of metabolites such as G3P, GSH, GSSG, NAD, or NADH, and attenuated the effect of high glucose levels on TCA cycle metabolites. Each of those experimental measurements is also seen in our model predictions, suggesting the implemented mechanism (reducing glucose transport) is a promising hypothesis.

Our model unexpectedly predicted that altering the *g6pd* reaction (simulating agrimony supplementation) led to an increase in reduced glutathione (GSH) levels in the cell. Glutathione is among the most well-studied natural antioxidants, capable of preventing cell damage incurred by reactive oxygen and nitrogen species [112–115]. This emergent and unanticipated prediction from our model supports the potential utility of agrimony supplementation among diabetes patients. It is also interesting that the model was able to provide confirmation to a hypothesis regarding the mechanism of action of agrimony; namely, we predict that agrimony

targets the *g6pd* reaction in the PPP. Increased flux through this reaction can cause increased levels of antioxidants, which reduce cellular inflammation and enable the β-cell to properly function [31, 116–119].

Increasing the rate of the *ak* reaction is predicted to decrease insulin secretion without substantially changing the metabolic network dynamics. This may be due to its involvement in the AMP-activated protein kinase signaling pathway, which was not included in our modeling effort but has been shown to influence insulin secretion. Targeting the *ak* reaction may be of particular clinical interest, as it can increase insulin production and reduce the hyperglycemic pressures experienced by patients without affecting the survival of the β-cell. An experimental drug, bis(adenosine)-5'-pentaphosphate, which targets the *ak* reaction, has previously been studied as a vasoconstrictor [26]. Our work suggests that it may be of use in diabetes, repurposed to increase insulin secretion. Future work can assess its viability as an anti-diabetic treatment strategy.

## 4.2 Study Limitations

The model captures the dynamics of pancreatic β-cell metabolism and can be applied to study clinically relevant interventions. However, we acknowledge some aspects of the computational model that can be improved up on future work. The model does not account for heterogeneity within a population of cells and does not consider the metabolic or paracrine interactions between β-cell and the other islet cells, such as α- or δ-cells. As β-cells can exhibit a different metabolism depending on interactions with other cells, this would be a relevant direction for future model expansion. The model is built based upon prior modeling efforts from β-cells and other cell types. We used experimental data from the INS1 832/13 cell line to make the reaction velocities specific to the pancreatic β-cell; however, the form of the rate equations can also be refined based on β-cell-specific data as they become available. We focused on central carbon metabolism, but there are additional pathways that could be included. For example, the degradation of free fatty acids is thought to impair insulin secretion in β-cell, and may be of particular interest in diabetes; this is an avenue for future research. More broadly, future iterations of this work may address these limitations.

## 5. Conclusions

We present a novel kinetic model that can effectively be used to study the dynamics of central carbon metabolism in pancreatic β-cells. The model goes beyond existing models and consists of key pathways and metabolites known to be important in GSIS. The model has been trained and validated with published data from the INS1 cell line. The model simulates the effects of metabolic perturbations, predicting the metabolite levels and flux distributions upon knockdown or upregulation of specific enzymatic reactions. We pair the kinetic model with a data-driven modeling approach, thereby linking intracellular metabolism to insulin secretion. The model is a promising step towards effectively using *in silico* techniques to generate novel insights into pancreatic β-cells. Thus, our work complements experimental studies and can be used to identify novel treatment strategies for diabetes.

## Supporting information

**S1 Table. Experimental data used for model training.**
(XLSX)

**S2 Table. Stoichiometric matrix.**
(XLSX)

**S3 Table. Multiple *t*-tests comparing predictions with experimental data.**
(XLSX)

**S4 Table. *In silico* metabolite fold changes following metabolic perturbations.**
(XLSX)

**S5 Table. Model parameter values, units, and sources.**
(XLSX)

**S1 Supporting Information. Supplementary figures.**
(PDF)

**S1 Text. Model equations.**
(DOCX)

# Acknowledgments

The authors thank members of the Finley research group and the Pancreatic Beta Cell Consortium (especially the Metabolomics sub-group) for helpful discussions. Computation for the work described in this paper was supported by the University of Southern California Center for Advanced Research Computing (https://www.carc.usc.edu/).

# Author Contributions

**Conceptualization:** Scott E. Fraser, Stacey D. Finley.

**Data curation:** Dongqing Zheng, Kate L. White, Nicholas A. Graham.

**Formal analysis:** Patrick E. Gelbach.

**Investigation:** Patrick E. Gelbach.

**Project administration:** Stacey D. Finley.

**Resources:** Stacey D. Finley.

**Writing – original draft:** Patrick E. Gelbach, Stacey D. Finley.

**Writing – review & editing:** Patrick E. Gelbach, Dongqing Zheng, Scott E. Fraser, Kate L. White, Nicholas A. Graham, Stacey D. Finley.

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
