## [Decision Letter · Decision Letter 0]

5 Jan 2022

Dear Dr. Finley,

Thank you very much for submitting your manuscript "Kinetic and data-driven modeling of pancreatic b-cell central carbon metabolism and insulin secretion" for consideration at PLOS Computational Biology.

As with all papers reviewed by the journal, your manuscript was reviewed by members of the editorial board and by several independent reviewers. In light of the reviews (below this email), we would like to invite the resubmission of a significantly-revised version that takes into account the reviewers' comments.

We cannot make any decision about publication until we have seen the revised manuscript and your response to the reviewers' comments. Your revised manuscript is also likely to be sent to reviewers for further evaluation.

Sincerely,

Vassily Hatzimanikatis

Associate Editor

PLOS Computational Biology

Kiran Patil

Deputy Editor

PLOS Computational Biology

Reviewer's Responses to Questions

**Comments to the Authors:**

Reviewer #1: Diabetes is a common chronic disease manifesting when not enough insulin is produced by the pancreatic beta-cells. Although several efforts have been made within the community to understand the metabolism of these cells in response to high glucose, up to date, b-cells metabolism is not well characterized. In this work, the authors developed a computational modeling workflow that includes a comprehensive model for beta-cells expanding existing knowledge, data integration, parameter identification, and model simulations. With this model, the authors were able to study the metabolic sensitivity of these cells to glucose and simulate the effect of used drugs. The analysis presented here can be used to guide experimental design and drug development. Therefore, the content of this manuscript is highly relevant to the field.

Major comments:

1.Given the importance of energy in the metabolism of beta-cells, the authors could comment why they have decided not to include the ETC reactions in their model.

2.In figure 1, it is mentioned that the reaction equations are given in the supplementary material; however, I could not find this information.

3.In section 2.2, the authors say that the model contains a total of 56 metabolites and 61 reactions. However, only 39 metabolites (12glycolysis + 7ppp + 20tca) and 45 reactions (13 glycolysis + 7 ppp + 25tca) are presented in the main text. It would be nice that the authors specify in the text which reactions were associated to the polyol pathway and also elaborate on the rest of the reactions that are part of the network.

4.In line 206, it is not clear why the model contains 92 reaction velocities (Vmax) if there are 61 reactions in the network. The authors can consider describing the type of the rest of the 385 parameters. Furthermore, the authors should explain what are the directionalities considered for the reactions.

5.Line 251, it is not clear to which parameters they are referring. If it is the case, the authors can explicitly write “correlated to their corresponding v_F”. It may be helpful to explain the nomenclature used for Vmax, v_F and v_R, which of them represent parameters, and which flux values.

6.In Figure 5, the authors write that all the Vmax are depicted, but there are only 33 reactions in the plot. Are these chosen reactions? In this graph, what does the red star next to some reaction names represent?

7.Line 539, in the figure, only the case of increased flux is shown.

8.Is there any evidence in the literature supporting the results for section 3.3? Even if it is partial, adding this would improve the impact of the model.

9.In figure S3, it is not clear why the flux through the reactions gssgr and gpx is increased but not the flux through ak.

10.The authors should consider creating a repository with the code and data necessary to ensure the reproducibility of the results of the paper. I could not find any reference to the code in the current manuscript.

Minor:

1.Line 232 typo “the”

2.Line 527-528 redundant “perturbing” “perturbed”

Reviewer #2: The manuscript by Gelbach and co-workers describes a computational model based on ordinary differential equations of the pancreatic beta cell carbon metabolism formulated as a tool to study the glucose stimulated insulin secretion. In general, the model is presented as very accurate although the reader has no means to judge or contrast this claim. In fact, the results presented are mainly qualitative in terms of biological significant information that can be contrasted with different sets of experimental data.

Model and data presentation:

On the one hand there is a careful and detailed description of the methods used for adjustment (model training) of parameters and sensitivity analysis of the main parameters such as Vmax. On the other hand, the authors do not present neither the model kinetic equations nor the code used in their algorithms. Also, no model simulated flux values or the so much advertised by the Authors flux distributions are shown, that could be contrasted with real fluxes in any model. For example, mitochondrial respiration measurements can be used to estimate the TCA cycle flux in cells, and be used to compare the simulated flux values. In such scenario, the usefulness of this manuscript is very limited and not a real contribution that could benefit the field. I was surprised by the plots in Figures 6 and 7 displaying only qualitative results in the form of colors of intermediates concentrations.

Another important variable value is related to the redox state in the various conditions. The authors mention increases in NADH, NAD and glutathione but fail to mention and compare the state of the redox couples (NADH/NAD). The importance of this metric is the abundance of biological data under different experimental condition that could be used to validate this model.

Given the objective of searching for new targets to treat (type 1) diabetes, the reliability of the biological assumptions behind the model equations and the goodness of the comparison of biological variables is fundamental. Without that possibility of contrasting the modeling results, the model is not useful.

Figures comments:

Figure 1: The color scheme chosen provides very bad contrast in several metabolites which added to the low resolution of the pdf document render this scheme rather unreadable. The lettering should be clearly contrasted with the background for readability.

Figure 2. The scale of ordinates is logarithmic. I understand that this is due to the orders of magnitude difference between the various metabolites. However, the error bars do not mean much in a logarithmic scale and the difference between computed and experimental data are downplayed with such a scale. Instead of showing the plot, I suggest adding a table with the real values together with standard errors. Alternatively, a plot in linear scale could give a more quantitative appreciation of the "goodness" of the simulation results with the computational algorithm.

Figure 3. The data are normalized. However, in this way the reader cannot compare the simulations to experimental data. On the one hand, no information about the units that are being used (specifically in Figure 3B). This is completely abstract presentation not appropriate for a journal with a large biological audience.

Figure 5: There is no indication of the color scale? Additionally, the presentation is rather problematic, since most representations display the "adjusted" variable in the x axis and the resulting (e.g. the change in metabolite level) in the y-axis, which is the opposite display in this figure.

Minor points:

Line 101: What do you mean by "no sufficient model"? Please, rephrase.

Line 232: typo “he” instead of “the”

Line 241: Clarify the origin of the “training” dataset: Spegel et al or Malmgren et al or both?

Line 249. Pairwise parameter identifiability analysis is jargon. Please, explain.

Line 677 Diabetes type 1 I guess is what the authors are referring to…

Line 705 glutathione, reduced or oxidized?

Line 730. Why is a model limitation not considering the heterogeneity of the lack of interactions with other cell types? Which would be the consequences of considering such interactions? Or the cellular heterogeneity?

Reviewer #3: Summary

The authors have constructed a kinetic model for central carbon metabolism in pancreatic beta cells using previously published intracellular metabolite concentration data. This kinetic model is then used to generate flux distributions, which are correlated with insulin secretion using a partial least-squares model. The authors have used this model to predict the pharmacological effects of commonly prescribed anti-diabetic drugs such as metformin and identify novel targets that can be modulated to enhance insulin secretion in the beta cells.

Major comments:

The authors have parameterized the kinetic model for central metabolism after identifying influential parameters using the eFAST method. The authors must parameterize the complete kinetic model using particle swarm optimization to justify their assumption that their neglected parameters are indeed not influential. This is important because the authors have identified distinct kinetic models that explain the metabolomics data to the same extent. Poor resolvability of models could also suggest data insufficiency.

The authors must perform a more rigorous statistical analysis that includes a χ^2-goodness of fit to demonstrate that the parameterized kinetic model is statistically significant.

The experimental data shown in figure 2 is plotted in logarithmic scale, suggesting the coefficient of variation in the experimental data is very large. The authors must comment on how this would impact the variability in flux predictions in their eight parameterized models.

The low Q^2 Y values reported during PLS have been attributed to the size of the dataset (only five time points). Because the model is constructed to modulate insulin secretion, additional data is needed to validate the interesting hypotheses generated in section 3.2. In particular, the authors must provide additional evidence to substantiate their associations between adenylate kinase and TCA cycle activity on insulin secretion.

It is surprising to note that perturbing the Vmax parameter of most reactions has little to no effect on the metabolites from pentose phosphate pathway and redox metabolism. The authors must explain why this occurs in the context of their kinetic model. Were the enzyme saturation levels too low due to a very large K_M?

Minor comments:

In line 82, the authors claim that “measurements of metabolite pool sizes do not give insight into the cellular dynamics or kinetics…”. Since fluxes are directly related to changes in the metabolome, the authors’ claim may not be accurate. It is recommended that the authors reword this claim to make a case for constructing a kinetic model.

**Have the authors made all data and (if applicable) computational code underlying the findings in their manuscript fully available?**

Reviewer #1: **No: **While all relevant data used in this study is explained in the manuscript and SI, I could not find any reference to where the code and data files to reproduce the results can be found.

Reviewer #2: **No: **Equations or code are not included. Only some of the metabolite data and simulated values is provided. No metabolic fluxes data.

Reviewer #3: None

PLOS authors have the option to publish the peer review history of their article (what does this mean?). If published, this will include your full peer review and any attached files.

Reviewer #1: No

Reviewer #2: No

Reviewer #3: No
---

## [Decision Letter · Decision Letter 1]

22 Apr 2022

Dear Dr. Finley,

Thank you very much for submitting your manuscript "Kinetic and data-driven modeling of pancreatic ß-cell central carbon metabolism and insulin secretion" for consideration at PLOS Computational Biology.

As with all papers reviewed by the journal, your manuscript was reviewed by members of the editorial board and by several independent reviewers. In light of the reviews (below this email), we would like to invite the resubmission of a significantly-revised version that takes into account the reviewers' comments.

We cannot make any decision about publication until we have seen the revised manuscript and your response to the reviewers' comments. Your revised manuscript is also likely to be sent to reviewers for further evaluation.

Sincerely,

Vassily Hatzimanikatis

Associate Editor

PLOS Computational Biology

Kiran Patil

Deputy Editor

PLOS Computational Biology

Reviewer's Responses to Questions

**Comments to the Authors:**

Reviewer #1: The authors have addressed all my comments.

Reviewer #2: I have evaluated the response of the authors to the specific points I raised with respect to the original manuscript. I found the model formulation and results rather disappointing both computationally and from a biological standpoint. The main issue is that this model is not able to evolve to a steady state, and the only quantitative data shown in Figure 2B really does not make any biological sense, likely because of an ill-designed model. There are no mass conservations relations. There are sinks and sources of intermediary metabolites not accounted for, or there is accumulation or depletion of metabolites that are not appropriately described in the manuscript. As described in the equations that the authors now added in response to my request to their first submission, this model won’t reach a steady state.

A steady state behavior is important for several reasons: 1) the need to know where in the parametric space the model behavior maps and to distinguish between steady and transient states; 2) from a steady state, the modeler can introduce a perturbation to know whether the model behavior returns to the state before the perturbation or if it evolves to another steady state, if not then the model is in an unstable region of the parametric state and this should be known to distinguish from transient states; 3) if using the model to simulate a flux distribution throughout a metabolic pathway, it should be known whether the parameterization makes any biological sense because, in principle, the space of solutions of a model is big and the key biological question is to which solution in that space belongs the experimental data that the model is trying to simulate.

In other words, metabolic networks in biological systems typically show features of stability of the internal medium (homeostasis/homeodynamics) and this should be reflected by the model’s steady state. It is also important to demonstrate its robustness against alterations in the environment which will again be reflected by the ability of the system to return to the original state after a perturbation. The condition of steady state will not show, of course, at the point of a perturbation (e.g. glucose concentration increase point from 2.8 to 16.7 mM), but the simulations of the metabolic behavior should come back to a steady state and the authors have failed to show a temporal evolution toward a steady state. Instead, they show averages of time points during the perturbation. I believe the authors need to improve the design of the model as applied to pancreatic beta-cells and use the model to answer meaningful questions.

Reviewer #3: The authors have done a very good job addressing all of my previous comments and I will be excited to see this work published

**Have the authors made all data and (if applicable) computational code underlying the findings in their manuscript fully available?**

Reviewer #1: Yes

Reviewer #2: **No: **The data about the fluxes and metabolite concentrations are hidden behind averages or meaningless color coded figures

Reviewer #3: None

PLOS authors have the option to publish the peer review history of their article (what does this mean?). If published, this will include your full peer review and any attached files.

Reviewer #1: No

Reviewer #2: No

Reviewer #3: No
---

## [Decision Letter · Decision Letter 2]

3 Aug 2022

Dear Dr. Finley,

Thank you very much for submitting your manuscript "Kinetic and data-driven modeling of pancreatic ß-cell central carbon metabolism and insulin secretion" for consideration at PLOS Computational Biology.

As with all papers reviewed by the journal, your manuscript was reviewed by members of the editorial board and by several independent reviewers. In light of the reviews (below this email), we would like to invite the resubmission of a significantly-revised version that takes into account the reviewers' comments.

We cannot make any decision about publication until we have seen the revised manuscript and your response to the reviewers' comments. Your revised manuscript is also likely to be sent to reviewers for further evaluation.

Sincerely,

Vassily Hatzimanikatis

Associate Editor

PLOS Computational Biology

Kiran Patil

Deputy Editor

PLOS Computational Biology

Reviewer's Responses to Questions

**Comments to the Authors:**

Reviewer #1: The authors had addressed all my comments.

Reviewer #2: After two rounds of evaluation of the manuscript by Gelbach et al. I can state that the main problem is the presentation of the model and the results. The authors have shown that the model is able to achieve a steady state at 16.7mM Glucose (but they have not shown that as well for the condition at 2.8 mM glucose). Also, they show through the stoichiometric matrix that the model is well balanced. However, there are still numerous issues that should still be clarified.

Concerning the model, each of the 65 enzyme rate expressions has many parameters, in addition to those 85 (although the authors state 81 in the main text) that need to be adjusted following the experimental data available (not for every metabolite) of 56 metabolites. Not only the value, but also the units and source of parameters should be stated in the supplementary information. This reviewer wonders if there is enough data to constraint such a large parameter sets or if the system is underdetermined? And what is the method used to constraint that large set of parameters?

It is not clear how the authors discriminate experimental data on metabolites that are present both in mitochondria and cytosol (shared metabolites). The authors state that there is only one pool of metabolites detected by MS, but figure 1 shows concomitant changes in model and experiment for most shared metabolites. Moreover, the conditions in which the comparison is made are completely obscure.

No parameters set(s) are defined to clarify how the simulations producing the results presented in each figure where performed. For example:

(a) Does the initial point at 2.8 mM Glucose before the perturbation correspond to a steady state? It should be at steady state when perturbed with 16.7 mM and then it should evolve in time reproducing the experimental data, does it?

(b) Moreover, the whole parametric set should be provided either in the main text or the supplemental information for a reader to be able to reproduce the results.

The figure legends are incomplete since the conditions in which the results shown were obtained are not stated.

The data in Figure 4 is specially challenging. The data about the fluxes show that most metabolites are far from a steady state, e.g. 100 (Units?) of glucose is being provided but only 25.1 glucose are consumed. About 15 (what?) of pyruvate are being consumed while less than 6 are entering the mitochondrial pyruvate pool. This way of presenting the fluxes is confusing and unorthodox, to say the least, mainly without the clarification of the conditions represented by these completely imbalanced fluxes.

In summary, I see that the authors have put a lot of emphasis in the statistical aspects of the modeling data while disregarding a correct presentation of the model with which they have performed that statistical analysis.

Reviewer #3: The authors have addressed all of our previous comments. We look forward to seeing this work published.

**Have the authors made all data and (if applicable) computational code underlying the findings in their manuscript fully available?**

Reviewer #1: None

Reviewer #2: **No: **Parameter units and sources thereof are lacking. Also adjusted parameter set is presented nowhere in the manuscript or supplement. Conditions to obtain the simulation results presented are missing as well.

Reviewer #3: Yes

PLOS authors have the option to publish the peer review history of their article (what does this mean?). If published, this will include your full peer review and any attached files.

Reviewer #1: No

Reviewer #2: No

Reviewer #3: No
---

## [Editor Report · Decision Letter 3]

8 Sep 2022

Dear Dr. Finley,

We are pleased to inform you that your manuscript 'Kinetic and data-driven modeling of pancreatic ß-cell central carbon metabolism and insulin secretion' has been provisionally accepted for publication in PLOS Computational Biology.

Best regards,

Vassily Hatzimanikatis,

Kiran Patil

PLOS Computational Biology

---

## [Editor Report · Acceptance letter]

7 Oct 2022

PCOMPBIOL-D-21-01847R3 

Kinetic and data-driven modeling of pancreatic ß-cell central carbon metabolism and insulin secretion

Dear Dr Finley,

I am pleased to inform you that your manuscript has been formally accepted for publication in PLOS Computational Biology. Your manuscript is now with our production department and you will be notified of the publication date in due course.

With kind regards,

Anita Estes
